# Exploring the vortex phase diagram of Bogoliubov-de Gennes disordered superconductors

**Bo Fan (范波)**[★] **and Antonio Miguel García-García**[†]

Shanghai Center for Complex Physics, School of Physics and Astronomy,
Shanghai Jiao Tong University, Shanghai 200240, China

★ bo.fan@sjtu.edu.cn , † amgg@sjtu.edu.cn

## Abstract

We study the interplay of vortices and disorder in a two-dimensional disordered superconductor at zero temperature described by the Bogoliubov-de Gennes (BdG) self-consistent formalism for lattices of sizes up to $100 \times 100$ where the magnetic flux is introduced by the Peierls substitution. We model substantially larger lattice size than in previous approaches ($\leq 36 \times 36$) which has allowed us to identify a rich phase diagram as a function of the magnetic flux and the disorder strength. For sufficiently weak disorder, and not too strong magnetic flux, we observe a slightly distorted Abrikosov triangular vortex lattice. An increase in the magnetic flux leads to an unexpected rectangular vortex lattice. A further increase in disorder, or flux, gradually destroys the lattice symmetry though strong vortex repulsion persists. An even stronger disorder leads to deformed single vortices with an inhomogeneous core. As the number of vortices increases, vortex overlap becomes more frequent. Finally, we show that global phase coherence is a feature of all these phases and that disorder enhances substantially the critical magnetic flux with respect to the clean limit with a maximum on the metallic side of the insulating transition.



# 1 Introduction

The application of a perpendicular magnetic field to a superconducting thin film leads to a very rich phenomenology. For type II superconductors at zero temperature, an Abrikosov lattice [1, 2] of vortices forms for intermediate fields. As temperature increases, topological defects, thermal vortices, starts to proliferate and eventually the lattice is melted through a Berezinskii-Kosterlitz-Thouless transition [3,4]. For lower temperatures, it has been identified theoretically, and later confirmed experimentally, an intermediate phase, termed an hexatic fluid [5–7] for lattices with hexagonal symmetry, that combines short-range positional order, like in a liquid, with a quasi-long-range orientational order as in the low temperature Abrikosov lattice phase.

The presence of disorder brings new interesting phenomena. A vortex tends to occupy regions where the order parameter is suppressed as a result of the disordered potential. At the same time, disorder pins vortices which prevents, or slows down, a dissipative response to a current, and therefore a finite resistivity. Deformations of a lattice of vortices, due to disorder, leads to the so called Bragg's glass [8–12] characterized by a power-law decay of the crystalline order so that some weakened form of diffraction peaks, and therefore discrete translational symmetry, coexists with glassy features. For a stronger disorder or field, a transition to a vortex glass [13–15] occurs characterized by both a relatively homogeneous repulsion among vortices in real space and, in Fourier space, a circular pattern [16] instead of sharp diffraction peaks that signal the complete loss of any discrete translational symmetry. A further increase in the disorder strength, or field, leads to either the loss of superconductivity or a fully disorder vortex phase where vortices repulsion is strongly suppressed.

A detailed experimental study [17], supported by numerical results based on the solution of the Bogoliubov-de Gennes (BdG) equation for small disordered lattices, revealed that vor-

tices occupy regions between superconducting islands which enhances phase fluctuations and eventually leads to a transition to a state formed by incoherent Cooper pairs. Translational symmetry of the vortex lattice seems to be lost even for a relatively weak disorder strength.

Transport properties in the presence of both disorder and magnetic field show rather unusual features. Experimentally, it has been observed an enormous increase of resistivity [18–20] for fields slightly above the one at which the insulating transition occurs. Surprisingly, a further increase of the magnetic field reduces the resistivity to values closer to the normal metal limit. The origin of these unexpected features is still under debate [21] though it is believed to be somehow related to residual correlations of the superconducting state [22–26] in the form, for instance, of localized phase-incoherent Cooper's pairs.

A more recent explanation of this phenomenon [27], based on an explicit numerical solution of the BdG equations in small lattices, is that there exists a region of magnetic flux strength where the conductivity still has a gap-like form for low frequencies but the superfluidity density vanishes. As a result, the resistivity becomes very large until larger magnetic fluxes close the gap completely.

Although disorder, temperature or magnetic field tend in general to suppress superconductivity, their combined effect can have a more complex behavior. For instance, as mentioned earlier, disorder hampers the motion of vortices, especially at low temperature, which suppresses dissipation and therefore potentially enhances superconductivity. Indeed, a recent study [28] of the XY model with a non-zero flux using Monte Carlo techniques [29–31] has found that disorder makes the superconducting state more robust against thermal effects. Similarly, disorder in certain circumstances can also enhance the superconducting critical temperature [32–39].

It is important to stress that, with a few exceptions Refs. [17, 27, 40–43] to be discussed later, theoretical research about vortices in disordered superconductors does not employ the microscopic and self-consistent BdG approach where the random potential is the one felt by the electrons that form the Cooper's pair. For instance, in the XY model, describing the phase dynamics, the Josephson couplings are random but they are not directly related to the random potential that model impurities in materials. Likewise, in stability studies of vortex lattices [8, 10–12, 44], disorder is just a random deviation of the vortex position from the one corresponding to a perfect Abrikosov lattice. In practical terms, this is qualitatively similar to the assumption that the disorder distribution of the impurities of the sample, typically Gaussian or box distributed, is also the one observed in the order parameter or other relevant observables of the superconducting state. However, this is not always the case.

There are substantial experimental [45, 46] and theoretical [32, 34, 37, 39, 47–57] evidence indicating that a microscopic approach is necessary to model quantum coherence effects, such as Anderson localization [58], induced by disorder that control the physics in certain region of parameters. This is especially true in two dimensions [59, 60] where even a weak disorder strength can trigger important localization [58] effects in the superconducting state. For instance, the amplitude of the order parameter becomes highly inhomogeneous [47, 48] in space with an emergent granular structure even on the metallic side of the superconductor-insulator transition. Close to the transition, the probability distribution of the order parameter amplitude is well described not by a Gaussian but by a broad log-normal distribution [49] and a parabolic $f(\alpha)$ spectrum [34, 37, 45, 46] typical of systems with multifractal-like features [51, 61, 62]. As mentioned earlier, a range of parameters has been identified where, due to this intricate spatial structure, the average order parameter and the critical temperature is enhanced by disorder [32, 35, 49]. The physical reason for this counterintuitive behavior is that although in many sites the order parameter is suppressed, in others it is substantially enhanced. We note that superconductivity does not require all sites to have phase coherence but only that a supercurrent can go through the sample. Recent experimental results [45, 46] are fully consistent with this theoretical picture.

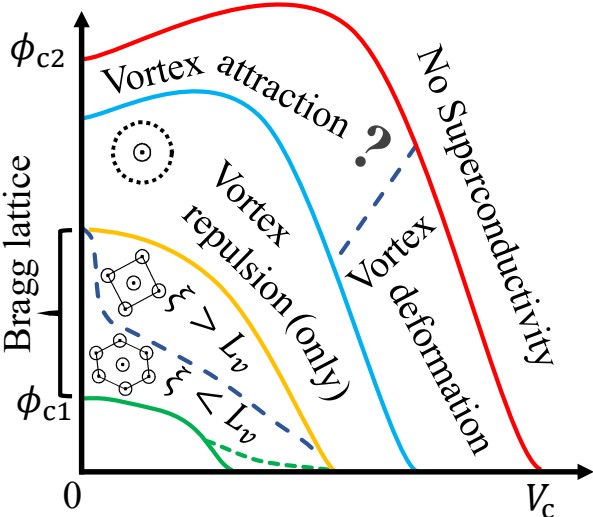

Figure 1: Vortex phase diagram. The cartoon summarizes the vortex distribution as a function of magnetic flux $\phi$ and disorder $V$. The red (green) line stands for $\phi_{c2}$ ($\phi_{c1}$) the upper (lower) critical magnetic flux as a function of the disorder strength $V$. In the region below the dashed green line, our results are not conclusive regarding the existence of a vortex lattice. Between the green and the sky blue line, we observe the expected Abrikosov triangular lattice. Upon increasing the magnetic flux, the lattice of vortices becomes rectangular when the average distance between vortices $L_v$ is smaller than the superconducting coherence length $\xi_0$. In the figure, we depict the Bragg lattice, the structure factor of the position of vortices. For larger fields, the circular pattern signals the vortex repulsion phase characterized by a loss of vortex translation symmetry, strong vortex repulsion and, on average, rotational symmetry of the position of vortices. In the phase termed vortex attraction, vortex repulsion is strongly suppressed and we observe strong vortex overlap in many cases. The question mark refers to the fact that we do not have a fully quantitative description of this phase. The phase termed *vortex deformation* is characterized by vortices with a vortex core that becomes spatially inhomogeneous and a highly deformed vortex profile. By no superconductivity, we refer to a region of vanishing superfluid density independently of its origin. We note that in the rest of regions, phase coherence holds. Indeed, disorder enhances the critical flux $\phi_{c2}$.

In view of that, a natural question to ask is to what extent the current picture of the effect of disorder in superconducting vortices, largely based on a phenomenological description of disorder, is modified if disorder is introduced microscopically and the calculation is carried out self-consistently.

In this paper, we employ the self-consistent BdG formalism [63,64] to address this problem in a two dimensional disordered superconductor in the presence of a magnetic flux, introduced by the so-called Peierls substitution. More specifically, we study quantitatively the vortex distribution as a function of disorder and flux strength, and also the spatial structure of a single vortex when the order parameter is sufficiently inhomogeneous. Moreover, we address the impact of disorder on global phase coherence and also in superconducting properties such as the average order parameter and the critical flux corresponding to the breaking of superconductivity.

The main results of this study are summarized in Fig. 1. For weak disorder, the most salient feature is an intermediate, in flux, rectangular vortex lattice phase in Fourier space between the expected triangular Abrikosov lattice at no or very weak disorder, and a phase characterized by

vortex repulsion but no translational order. When the magnetic flux is large enough, the vortex overlaps and there are signs of incipient frustration in the phase of the superconductor, see Fig. 18 in Appendix B. For sufficiently strong disorder, this phase melts into a disordered vortex phase, termed *vortex deformation* in Fig. 1, where the vortex core is spatially inhomogeneous with a highly deformed external profile, see Figs. 10 and 11. Another intriguing finding is that all of the above vortex distributions coexist with phase coherence. Even more, disorder enhances the critical magnetic field, see Fig. 14, especially on the metallic side of the transition where the average order parameter is enhanced, see Fig. 15(b), as well. The latter features are potentially relevant for design optimizations of superconducting devices for technological applications.

As was mentioned previously, the interplay of vortices and disorder using the BdG formalism has already been investigated in Refs. [17, 27, 40–43] but for substantially smaller sizes: at most $36 \times 36$ in those papers versus $100 \times 100$ in our paper. Moreover, these works do not address the two main problems studied in this paper: the change in the vortex lattice distribution as a function of disorder and magnetic flux and the spatial deformation and inhomogeneity of single vortices for strong disorder on the metallic side of the transition. The reason for that is of technical nature, the study of vortex lattices requires larger lattice size. Likewise, the convergence of the code slows down substantially in the strong disorder region which therefore requires both substantial computational resources together with the use of state of art numerical techniques.

Finally, we would also like to mention a recent study [28], see also Ref. [65], that considered the interplay of disorder, magnetic field and temperature by using an effective XY model for the phase of the order parameter. It was found that disorder enhances the robustness of the superconducting state against magnetic effects at finite temperature. However, the dependence of the vortex lattice with the disorder strength, the main focus of this paper, is not addressed. Moreover, quantum coherence effects are lost in this type of phenomenological approach. Therefore, there is no sizable overlap between our results and previous literature on this problem.

We start our study with an introduction of the model and the employed numerical techniques.

## 2 Model and method

The disordered superconductor is modeled by an attractive Hubbard model,

$$H = \sum_{ij\sigma} -t c_{i\sigma}^{\dagger} c_{j\sigma} + U \sum_{i} n_{i\uparrow} n_{i\downarrow} + \sum_{i\sigma} V_i n_{i\sigma} \,. \tag{1}$$

The effect of a perpendicular magnetic field $\boldsymbol{B}(r,t) = \nabla \times \boldsymbol{A}(r,t)$ is introduced by the so called Peierls' substitution $t \to t_{ij} = t \exp(i\phi_{ij})$ where $\phi_{ij} = \frac{\pi}{\phi_0} \int_{r_j}^{r_i} \boldsymbol{A}(r) d\boldsymbol{r}$ and $\phi_0 = hc/2e$ is the superconducting quantum flux. The magnetic field $\boldsymbol{B}$ is then given in terms of the flux $\phi$ in units of $\phi_0$.

We note that when $U = 0$ and $V_i = 0$, and for certain values of the magnetic field and the lattice size, so the flux becomes increasingly incommensurate, the Hamiltonian reduces to the celebrated Harper-Hofstadter model [66, 67] which displays an intricate band structure that becomes multifractal at the transition. In this study, since we are interested on vortex physics, we only focus on integer fluxes that are far from the Harper-Hofstadter limit even for no disorder. Likewise, recent reports [68, 69] on Hofstadter superconductivity are for clean systems and for a strength of the magnetic field much larger than the one considered in the paper. Therefore, there is no overlap with our results.

Returning to Eq. (1), in order to simplify the numerical calculation, the perpendicular magnetic field is chosen to be a time independent uniform field $\boldsymbol{B} = (0, 0, B_0)$. We can use the vector potential $\boldsymbol{A} = (-B_0 y, 0, 0)$ in the Landau gauge. We neglect the coupling of the magnetic field to the spin, the Zeeman term, as the field strength in our problem is relatively weak so the spin-splitting is too small to cause any substantial effect in the vortex physics in superconductors we are interesting in.

By performing a Bogoliubov transformation $c_{i\sigma} = \sum_n \left[ u_n(i)\gamma_{n\sigma} - \sigma v_n^*(i)\gamma_{n\tilde{\sigma}}^\dagger \right]$, where $\gamma_{n\sigma}$ and $\gamma_{n\sigma}^\dagger$ are fermion operators, we obtain the two dimensional Bogoliubov de-Gennes equations [48, 63, 64, 70] in the presence of a magnetic flux,

$$\begin{pmatrix} \hat{K} & \hat{\Delta} \\ \hat{\Delta}^* & -\hat{K}^* \end{pmatrix} \begin{pmatrix} u_n(r_i) \\ v_n(r_i) \end{pmatrix} = E_n \begin{pmatrix} u_n(r_i) \\ v_n(r_i) \end{pmatrix}, \tag{2}$$

where

$$\hat{K} u_n(r_i) = -t_{ij} \sum_\delta u_n(r_i + \delta) + (V_i - \mu_i) u_n(r_i), \tag{3}$$

and the sum $\delta$ is restricted to the four nearest neighboring sites. In our calculation, for simplicity, we use $t = 1$ as the unit of energy, and the superconducting flux quantum is $\phi_0 = \pi$. $V_i$ are random variables from an uniform distribution between $[-V, V]$. The local chemical potential including the Hartree shift is $\mu_i = \mu + |U|n(r_i)/2$, $\hat{\Delta} u_n(r_i) = \Delta(r_i) u_n(r_i)$, and the same definition applies to $v_n(r_i)$. The BdG equations are completed by the self-consistency conditions for the site dependent order parameter $\Delta(r_i) = |U| \sum_n u_n(r_i) v_n^*(r_i)$ and the density $n(r_i) = 2 \sum_n |v_n(r_i)|^2$. The order parameter can be written as $\Delta(r_i) = |\Delta(r_i)| e^{i\theta_i}$ where the non trivial phase $\theta_i$ in this mean field formalism is a direct consequence of the existence of a magnetic flux. The averaged charge density $\langle n \rangle = \sum_i n(r_i)/N$ is fixed by tuning the chemical potential $\mu$ at each iteration step.

Imposing the self-consistent condition, we solve eq. (2) numerically on a square lattice ($N = L \times L$). In order to minimize finite size effects, it is important to employ periodic boundary conditions at zero temperature. However, this is challenging due to the presence of the flux leading to a vortex lattice and the requirement of magnetic translation symmetry [71, 72]. Following previous literature [43, 71, 72], we have found that the optimal choice that minimizes finite size effects and respects magnetic translation symmetry, is the so called twisted boundary condition along $y$-direction $u_n(r_x, r_y + L) = \exp(i\pi r_x L \phi/\phi_0) u_n(r_x, r_y)$ and $v_n(r_x, r_y + L) = \exp(-i\pi r_x L \phi/\phi_0) v_n(r_x, r_y)$, where $r_x$ and $r_y$ are the lattice sites along $x$ and $y$ directions respectively.

It is important to stress that exact periodic boundary conditions can be imposed in the limit of no disorder where the amplitude of the order parameter is constant except in the vortex core where it vanishes. This is achieved by performing a singular gauge transformation [40, 73] so that the phase factor of the order parameter vanishes but a new phase factor appears in the hopping terms which makes possible to impose strictly periodic boundary conditions respecting at the same time magnetic translation symmetry. However, it assumes a constant order parameter amplitude so the solution is not self-consistent. In the very weak disordered regime, it can still be a good approximation because deviations from an Abrikosov lattice are small and can be accounted phenomenologically [73] by assuming a small random displacement of the vortex position. However, this approach completely breaks down for stronger disorder, especially around the insulating transition. Since we are mostly interested in the impact of disorder on vortices for a broad range of disorder strengths, we cannot adopt this exact periodic boundary condition scheme.

As a consequence, we could not find a way to exactly impose periodic boundary conditions because, unlike previous studies in the literature, we aim to keep the treatment of the amplitude and the phase of superconducting order parameter on equal footing which requires a

self-consistent treatment of the former so that we can study changes in the vortex profile due to disorder. At the same time, sizes must be as large as possible in order to do any quantitative analysis of the vortex lattice which prevent us using Dirichlet boundary condition. This constraints led us to choose the mentioned twisted boundary conditions along one direction. Additional technical details about the choice and implementation of boundary conditions in the presence of a magnetic flux are found in Appendix. A.

Another important technical issue that also requires a detailed description is the method to determine the position of the vortex. A vortex occurs in a certain region of the sample if the sum of the phase difference between two neighboring sites ($\theta_{i+\delta} - \theta_i$) in a closed path $\mathcal{L}$ is $2\pi$, namely, $\sum_{\mathcal{L}}(\theta_{i+\delta} - \theta_i) = \pm 2\pi$. The vortex core is then located at the center of the closed path. The inset of Fig. 6(b) shows the precise relation of the phase $\theta_i$ (red arrow) and the vortex core (red circle). Further details on the definition of a closed path and a vortex core are found in Appendix. B. Moreover, in order for the phase $\theta_i$ of the superconducting order parameter to be single-valued everywhere, $\phi/\phi_0$ must be an even number [40, 74], so that the accumulated phase difference $\sum_{\mathcal{L}}(\theta_{i+\delta} - \theta_i)$ along any closed path that contains a set of vortices is $2n_v\pi$, where $n_v = \pm 1, \pm 2, \cdots$. The findings of Section. 5 and Appendix. E confirm that for a satisfactory description of the vortex profile, especially in the strong disorder region, it is necessary the self-consistent calculation of the amplitude of the order parameter.

Finally, we comment on the choice of parameters in this study. We adjust the chemical potential so that the charge density is fixed at $\langle n \rangle = 0.875$ which is commonly used in the study of disordered superconductors [27, 37, 47, 48]. For the study of the vortex lattice, we set $N = 60 \times 60$ and coupling constant $|U| = 1.25$. With these parameters, the coherence length $\xi_0 \sim 12$ so the radius of the vortex $r_0 \sim 12$ will be similar which allows us to reproduce the Abrikosov lattice even in the clean limit. In some cases, we also study larger system size $N = 100 \times 100$ to confirm the results. When we study the vortex profile and spatial inhomogeneity within the vortex region, our focus shifts to a system with weaker coupling $|U| = 1$, leading to larger vortex, $r_0 \sim 15$, which helps us investigate in more detail the spatial structure inside the vortex. In order to get rid of the interactions between vortices, which might cause unexpected effects, we study only two vortices in such system with size $N = 60 \times 120$. In summary, with this choice of parameters, we are able to explore vortex physics, both for a single vortex and for a lattice of vortices, in the presence of disorder approaching the experimental region of weakly coupled conventional metallic superconductors.

## 3   Distribution of vortices in clean BdG superconductors: Abrikosov triangular lattice

In this section, we study the distribution of vortices as a function of the magnetic flux strength in the limit of no disorder where we expect to recover the Abrikosov triangular lattice solution originally obtained [1] from the phenomenological Ginzburg-Landau formalism.

In the clean limit, $V = 0$, the application of a sufficiently strong magnetic flux results in the creation of the Abrikosov lattice [1], a triangular lattice of vortices. The vortex distribution depicted in Fig. 2, for a $100 \times 100$ lattice, shows excellent agreement with an Abrikosov lattice in both real space and Fourier space.

We subsequently study the sample with different aspect ratios. Results depicted in Fig. 3 show that the triangular Abrikosov lattice is well reproduced, although in some cases, the Abrikosov lattice is stretched or compressed due to the shape. In Fourier space, we observed the expected sharp hexagonal Bragg pattern related to the triangular lattice in real space. Results for different sizes and aspect ratio, confirm the triangular Abrikosov lattice in the limit of no disorder.

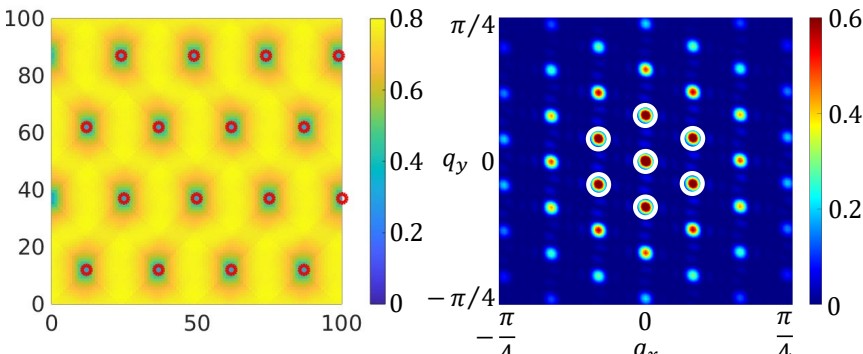

Figure 2: Left: The spatial distribution of the order parameter $|\Delta(r)|$ normalized by $\Delta_0 = 0.0894t$, which is the superconducting gap in the absence of disorder and magnetic flux. The position of vortices is marked by red circles. The Abrikosov triangular lattice is clearly observed. Right: The corresponding structure factor of the position of vortices. We obtain sharp Bragg peaks, marked by white circles in the figure, that correspond to the expected triangular Abrikosov lattice. This is consistent with the distribution of vortices in real space. The system size is $N = 100 \times 100$, and the magnetic flux $\phi/\phi_0 = 16$. The other parameters are $|U| = 1.25, \langle n \rangle = 0.875$.

# 4 Distribution of vortices in disordered superconductors

We now turn to the role of disorder in the vortex distribution at zero temperature. In Fig. 4, we depict the spatial dependence of the order parameter, resulting from the solution of the BdG equations, for different disorder strengths $V$ and magnetic fluxes $\phi/\phi_0$. Red circles stand for the vortex position. For no magnetic field, the order parameter clearly develops an intricate spatial structure as disorder increases which is starkly different from the box distribution of the disordered potential. Indeed, for $1 \gtrsim V \gtrsim 2$, rather conclusive analytic [33], numerical [37] and experimental [46] evidence indicates that the spatial distribution of the order parameter follows a broad log-normal distribution. Therefore, phenomenological approaches to the study of vortices in disordered superconductors that assume that implicitly assumes that the order parameter spatial distribution is that of the disordered potential can only be quantitatively correct in the very week disorder limit $V \leq 0.5$ where quantum coherence effects induced by disorder are not important.

## 4.1 Weak disorder region

In the weak disorder region $V = 0.5$, the distribution of vortices is rather sensitive to $\phi/\phi_0$. For $10 \leq \phi/\phi_0 \leq 16$, we still observe clear regularities that points to a deformed triangular Abrikosov lattice. However, a larger $\phi/\phi_0$ induces larger spatial inhomogeneities in the order parameter that translates into a more complicated vortex pattern. It seems that it becomes energetically favorable that vortices occupy regions where the order parameter is suppressed. We note that, in two dimensions, the effect of sufficiently strong disorder induces incipient quantum localization effects such a log-normal spatial distribution of the order parameter [33,34]. However, a disorder strength $V = 0.5$ is too weak to cause any significant localization effect. The distribution of probability is indeed still close to Gaussian, see appendix G, though with a comparatively larger standard deviation.

For a larger field $\phi/\phi_0 = 24$, the vortex positions do not seem to follow any pattern. For larger field $\phi/\phi_0 = 36$ and $64$, we cannot discern vortices clearly because strong overlap in some cases which we think indicates that this must be close or above the critical field at which the loss of superconductivity takes place.

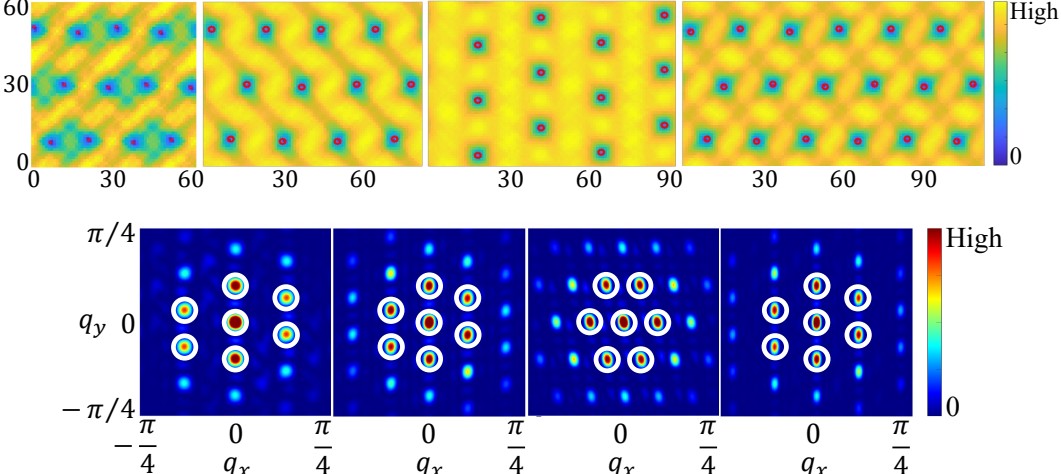

Figure 3: Top panel: The spatial distribution of the order parameter $|\Delta(r)|$ normalized by $\Delta_0 = 0.0894t$, and the vortices core (red circles). The height of the sample is fixed at 60, while the length of the sample are 60, 80, 90 and 110 from left to right. The Abrikosov triangular lattice is clearly observed though for 60 is slightly compressed. The magnetic flux is $\phi/\phi_0 = 12$ for 60, 80, 90, and $\phi/\phi_0 = 18$ for 110. Below is the corresponding structure factor. We obtain sharp Bragg peaks, marked by white circles in the figure, that correspond to the expected triangular Abrikosov lattice. The other parameters are $|U| = 1.25, \langle n \rangle = 0.875$.

In any case, the spatial distribution of the order parameter is not enough for a quantitative description of the vortex distribution. For that purpose, we compute next the structure factor with respect to the position of the vortices [7, 16, 75]. We note that lattice symmetries in real space can be characterized by the pattern of Bragg peaks in the first Brillouin zone.

### 4.1.1 From triangular to rectangular vortex lattice in Fourier space

We now provide a more quantitative analysis of the nature for the vortex lattice as a function of disorder and magnetic flux by the Fourier transform with respect to the vortex positions. For $V = 0.5$ and $\phi/\phi_0 \leq 18$, we observe, see Fig. 5, an hexagonal structure in Fourier space which is a signature of the triangular Abrikosov lattice. Unexpectedly, around $\phi/\phi_0 \sim 20$, the hexagonal lattice in Fourier space transforms into a rectangular lattice.

We first analyze the difference of the maximum angle $\theta_x$ and minimum angle $\theta_n$ of the triangle formed by three neighboring vortices, and the distance between two vortices, as a function of the magnetic flux. The results are shown in Fig. 6. In the triangular Abrikosov lattice, $\theta_x = \theta_n = \pi/3$, which leads to $\cos(\theta_x - \theta_n) = 1$. For a right triangle, $\theta_x = \pi/2$ and $\theta_n = \pi/4$, which leads to $\cos(\theta_x - \theta_n) = \sqrt{2}/2 \sim 0.7$. Those features are well captured in Fig. 6(a) that shows the transformation of the vortex distribution from a triangular lattice to a rectangular lattice.

A distinct feature of the rectangular lattice is that the distance between vortices is smaller than the typical vortex separation $L_v$, which is determined by the coherence length $\xi_0 = 12$, obtained in the clean limit, so vortex overlap. This overlap is energetically unfavorable in the clean case. However, disorder may make it possible because vortices gain energy in locations where the order parameter is suppressed. Therefore, the observed rectangular distribution is a compromise between disorder that tend to group vortices with no spatial symmetry and magnetic flux that tend to a more symmetric triangular vortex distribution.

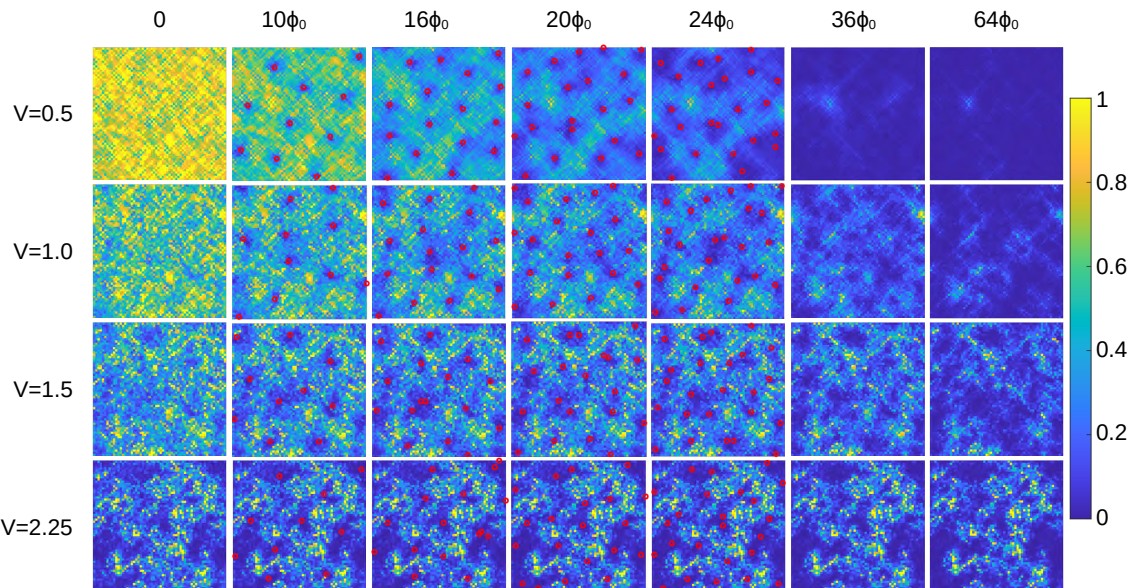

Figure 4: The spatial distribution of the order parameter $|\Delta(r)|$ normalized by $\Delta_0 = 0.0894t$ with $|U| = 1.25$, $\langle n \rangle = 0.875$. The position of vortices is represented by red circles. Disorder strength is $V = 0.5, 1.0, 1.5$ and $2.25$ in units of the hopping energy from top to bottom. The magnetic flux strength is, from left to right, $\phi/\phi_0 = 0, 10, 16, 20, 24, 36$ and $64$. By increasing disorder, the spatial distribution of the order parameter becomes strongly inhomogeneous. As expected, an increasing magnetic flux, suppresses the order parameter which effectively becomes more inhomogeneous. In the region of strong magnetic flux ($\phi/\phi_0 \geq 36$), close or at the transition, we do not mark the vortex position because, see section 5 and Appendix. A, vortex overlap and single vortices are deformed especially in the strong disorder region which makes difficult to determine its location.

More specifically, the differences between the non-overlapping vortices and the overlapping vortices are illustrated in the inset of Fig. 6(b) where a clear deformation is observed of the vortex arrow, with respect to that of a single isolated vortex, in the region between the two vortices.

These results further support that vortex overlap plays an important role in the triangular to rectangular lattice transition. In order to fully confirm the existence of this intriguing rectangular phase, we repeat the analysis for a larger sample size $L = 100$ in Appendix. F. A larger size leads to a larger number of vortices which makes the Fourier analysis much more accurate. The observation of a sharp rectangular pattern in Fourier space for $L = 100$, see Appendix. F, provides strong evidence of the existence of a rectangular vortex lattice in real space and sufficiently weak disorder far from the critical region.

We note that a similar transition from a hexagonal vortex lattice to a rectangular vortex lattice in Fourier space is also observed in FeSe [75] and LiFeSe [16]. In these experiments, the transformation is attributed to vortex overlap. A direct comparison with our results is not possible because these iron-based materials are multi-band superconductors. The order parameter is thus expected to have a non trivial angular dependence. By contrast, our model is disordered, single-band and the order parameter has s-wave symmetry and therefore no angular dependence.

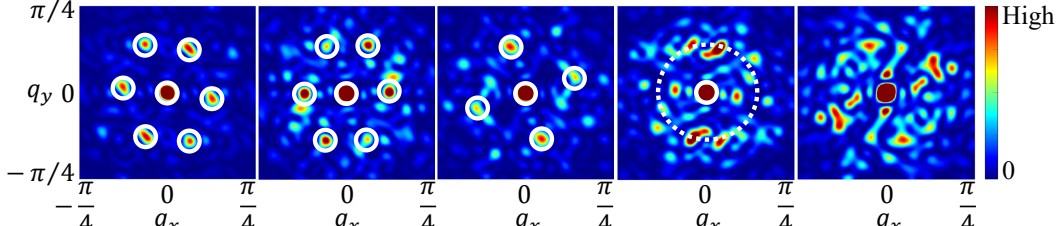

Figure 5: Structure factor of the position of vortices in the weak disorder region $V = 0.5$. The magnetic flux is $\phi/\phi_0$ is $16, 18, 20, 22$ and $24$ from left to right. For $\phi/\phi_0 = 16, 18$, the pattern is consistent with a triangular lattice in real space. However, a small increase, $\phi/\phi_0 = 20$, leads to a transition to a rectangular lattice. An increase in the flux ( $\phi/\phi_0 = 22$) results in a close to circular pattern signaling the absence of translational symmetry. Physically, a circular pattern indicates a combination of vortices repulsion and the restoration of rotational symmetry [65, 76]. This circular pattern eventually disappears for $\phi/\phi_0 \geq 24$ which signals a fully disordered phase with no clear vortex repulsion. To make the pattern more evident, we have set some cut-off value, referred by High in the plot.

### 4.1.2 From rectangular vortex lattice to vortex repulsion and beyond

By a further increase of the magnetic flux $\phi/\phi_0 \geq 22$, the peaks that characterize the rectangular lattice phase become gradually smeared out. Some structure, closer to a circle, remains which signals vortex repulsion but loss of any discrete translational symmetry and the on average restoration of rotational symmetry of the position of vortices. Therefore, although the inhomogeneity of the order parameter destroys any lattice structure in Fourier space, the magnetic flux still maintain vortices well separated. This gradual destruction of discrete translational symmetry has been observed [76] experimentally.

For $\phi/\phi_0 \geq 24$, no clear structure can be discerned in Fourier space which is typically associated with a vortex disordered phase where vortex repulsion is gradually weakened. In this weak disorder region, with no multifractal effects, we expect rotational symmetry to still continue in the region close to the transition. However, larger lattices, with a larger number of vortices, leading to a sharper pattern in Fourier space, are necessary for a full characterization of this phase. More specifically, it would be interesting to determine whether a clear diffraction disk, characteristic of rotational symmetry in the position of vortices, is observed.

## 4.2 Strong disorder region

For stronger disorder ($V \sim 1.5$), see Fig. 4, and not too strong fields, the system is still superconducting, see next section, but the order parameter has large spatial inhomogeneities and vortices tend to be located in regions where the order parameter is heavily suppressed. Moreover, vortices become spatially inhomogeneous. After averaging over different vortex cores, which smooths out inhomogeneities, the averaged vortex profile is still quite sensitive to the disorder strength, see Fig. 10 and Appendix E for more details. A Fourier analysis, see Fig. 7, confirms this point. The observed circular pattern for $\phi/\phi_0 \leq 24$, suggest, as in the weak disordered region, the restoration, on average, of the rotational symmetry, the breaking of any remnants of discrete translational invariance, and the persistence of strong vortex repulsion.

The circular pattern finally disappears for $\phi/\phi_0 \geq 28$. In this region of stronger fields, it is unclear whether rotational symmetry is restored because multifractal-like properties of the order parameter distribution may control completely the position of vortices. In any case, the system size is not large enough to provide a more quantitative characterization.

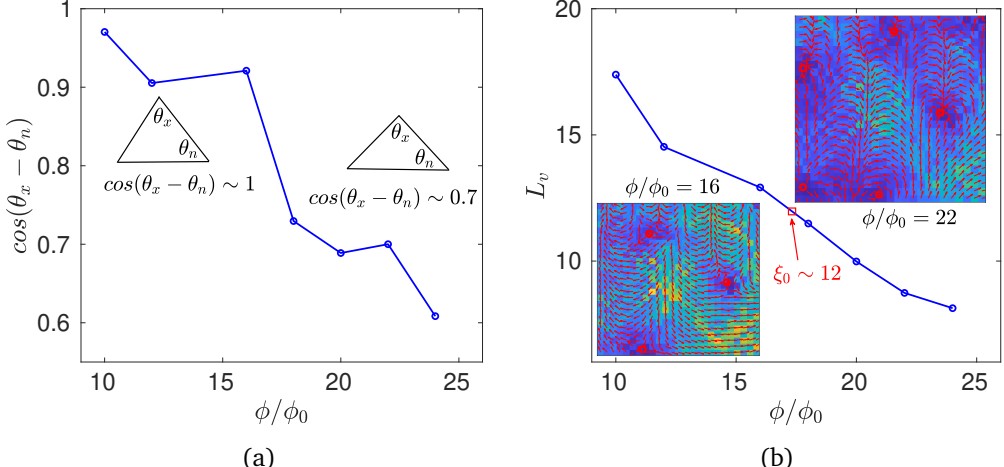

Figure 6: (a) The cosine of the differences of the maximum angle $\theta_x$ and the minimum angle $\theta_n$ in the triangle formed by three vortices. By increasing the magnetic flux, there is a transition from equilateral triangle to right triangle. (b) The vortex lattice spacing $L_v$ as a function of the magnetic flux. Inset: spatial distribution of the order parameter (color code of Fig. 4) for two different magnetic fluxes including the extra phase (red arrow) due to the magnetic flux and the vortex core (red circle). The size of the vortex lattice spacing in the clean limit is close to the coherence length, $\xi_0 \sim 12$, so we expect that smaller spacing will lead to vortex overlap.

For $V \gtrsim 1.5$, which is close to the transition, we find no trace of vortex lattice or glass structure. The position of vortices seem to be dictated by the sample regions where the order parameter has an especially small value. Therefore, no vortex repulsion is observed. Moreover, the inhomogeneities inside the vortex core becomes even stronger, see Fig. 11. For these reasons, in many cases, it becomes increasing difficult to precisely determine the position of isolated vortices.

For a sufficiently strong field, we expect that the vortex positions are ultimately controlled by the multifractal-like properties of the order parameter. However, as mentioned above, we could not find a precise characterization that would allow a more quantitative description of the vortex distribution. Larger sizes and more vortices would be necessary for that purpose.

A general feature of the strong disorder region is the relative insensitivity of the order parameter to the increase of the magnetic flux. In part, this is due to the fact that we model the magnetic flux in the so called Peierls substitution that neglects the coupling of the spin to the magnetic field. However, it also contributes that vortices occur in regions where the superconducting order parameter is already heavily suppressed by disorder, which is amplified by coherence effects, so they barely induce a further suppression. We shall see in Section 6 that this feature has important consequences in observables such as the critical magnetic flux or the spatial average of the order parameter. Finally, we note that in this section we have employed a $60 \times 60$ lattice. The reason for that is twofold, on the one hand numerical convergence is much slower as disorder increases. On the other hand, finite size effects are suppressed by disorder so, at least in the region where disorder destroys the vortex lattice. We do not think larger sizes $> 60 \times 60$ will change the results qualitatively. However, as mentioned earlier, larger lattices will be necessary for a more quantitatively description of the vortex distribution in this region.

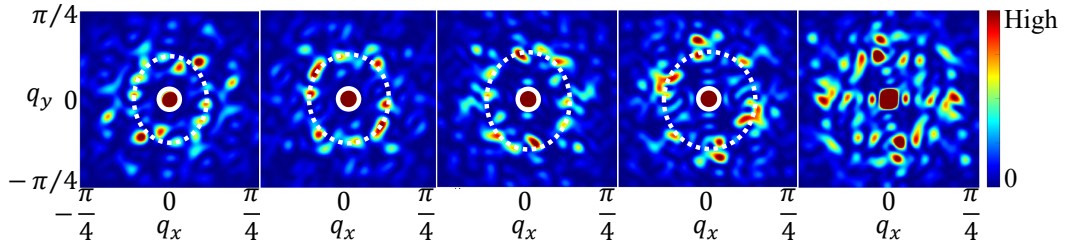

Figure 7: Structure factor of the position of vortices in the strong disorder region $V = 1.5$. The magnetic flux is $\phi/\phi_0 = 10, 16, 20, 24, 28$ from left to right and the lattice size is $60 \times 60$. Unlike the weak disorder $V \sim 0.5$ region, we do not observe the triangular or rectangular lattice phases. The spatial distribution of the order parameter is too inhomogeneous for the formation of any form of vortex lattice. For $\phi/\phi_0 \leq 24$, the system is characterized by vortices that still repel each other but have lost discrete lattice symmetry. For larger fields, vortex repulsion is strongly weakened. The vortex distribution approaches the fully disordered phase.

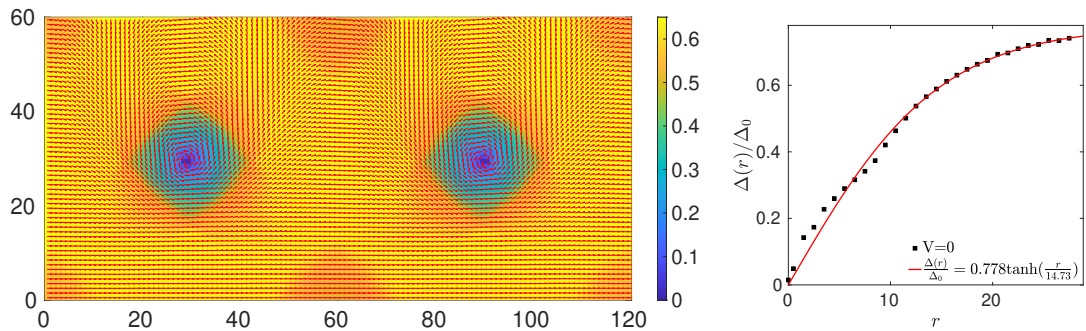

Figure 8: Left: The spatial distribution of the order parameter and its phase in the clean limit when there are four vortices. Right: The profile of the order parameter for the vortex. The red solid line is the fit to the Ginzburg-Landau prediction Eq. 4. The other parameters are $|U| = 1.0$, $\langle n \rangle = 0.875$.

## 5 The vortex profile

In the previous section, we showed that vortices occur in regions with heavily suppressed superconductivity. One natural question to ask is whether the vortex shape is sensitive to the spatial distribution of the superconducting order parameter. It is also important to explore the profile of single vortices in the presence of disorder, so that we can have a better understanding of the interplay of disorder and magnetic flux.

According to the Ginzburg-Landau (GL) theory [77, 78], the profile of the order parameter inside a vortex neglecting disorder effects is:

$$\frac{\Delta(r)}{\Delta_0} = a_s \tanh(r/r_0), \tag{4}$$

where $a_s \Delta_0$ is the spatial average of the order parameter in the absence of magnetic field and $r_0$ characterizes the vortex size. In the clean limit, $r_0$ is similar to the superconducting coherence length $\xi_0$. In the inhomogeneous case, to have smoother results, we obtain the $\Delta(r)/\Delta_0$ by averaging over points in the vortex core at the same distance of the center. The results with only two vortices are illustrated in Fig. 8 ∼ 11. In the clean and weak disorder limit, the results fit well with the GL prediction. Additional results are presented in Appendix E. The best fitting for $r_0$ in the weak disorder region $V = 0.5$ is $10 \leq r_0 \leq 14$, which is slightly smaller

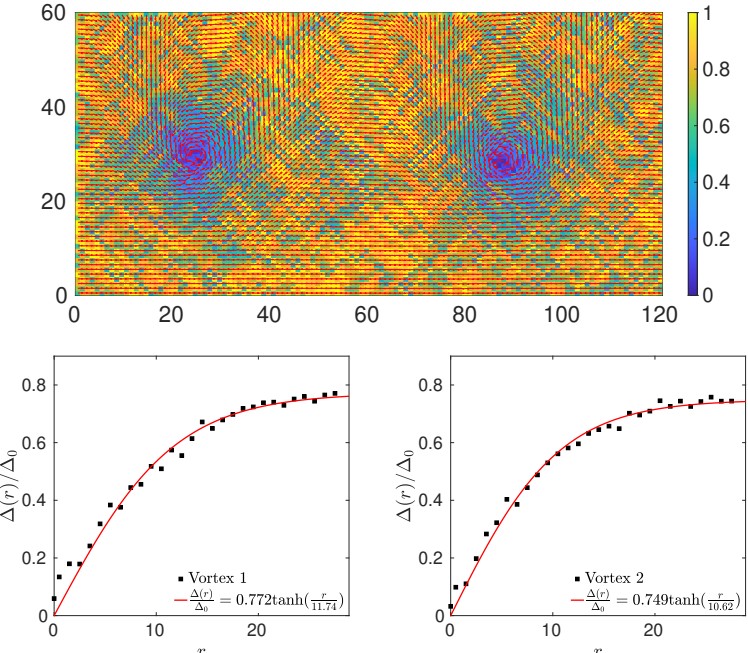

Figure 9: Upper: The spatial distribution of the order parameter amplitude and phase in the presence of a weak disorder strength $V = 0.5$. Lower: the corresponding vortex profile. The red solid line is the fit to the Ginzburg-Landau Eq. 4 prediction. Vortex 1 (2) stands for the left (right) vortex of the upper plot. Other parameters are $\langle n \rangle = 0.875$ and a lattice size $60 \times 120$. We employ a weaker coupling $|U| = 1.0$ so that the vortex is larger which facilitates the study of its core and profile.

than $r_0 = 14.73$ in the clean limit. However, when the disorder is stronger, the fittings become much worse, and the fitted parameter $r_0$ varies in a much larger region because the vortex profile is no longer circular. This is directly related to the fact that the spatial distribution of the order parameter is dominated by disorder which in this region is highly inhomogeneous.

More specifically, if we define the vortex profile by the spatial distribution of the phase in the vortices region, shown in Fig. 10 and 11, the vortices have the shape of heavily suppressed superconducting order parameter region which is far from circular but rather elongated and with no apparent symmetry. We are not aware of any previous research about this intriguing phase characterized by deformed vortices with inhomogeneous vortex cores.

Moreover, these results in the intermediate and strong disorder hint that in the weak $|U| \leq 1$ coupling limit, it may be possible to observe multifractal vortices [33, 37, 45, 46], namely, vortices whose shape is directly influenced by the multifractal-like features of the spatial distribution of the order parameter.

Finally, we note that by averaging over different vortices we recover an approximate circular shape, as is shown in Fig. 23 of Appendix E. In the presence of a stronger disorder $V = 2.25$, even identifying the vortex core is problematic and therefore we can not perform an average over the vortex cores. In this limit, which is around the superconductor-insulator transition, as expected, the vortex core deviate strongly from the GL prediction and the spatial structure is highly inhomogeneous, the outer profile seems to be very sensitive to the details of the disorder potential but highly elongated in general. Intriguingly, the phase of the order parameter seems to form the so-called Josephson vortex [27] that needs to be defined over a rather long and non-circular path.

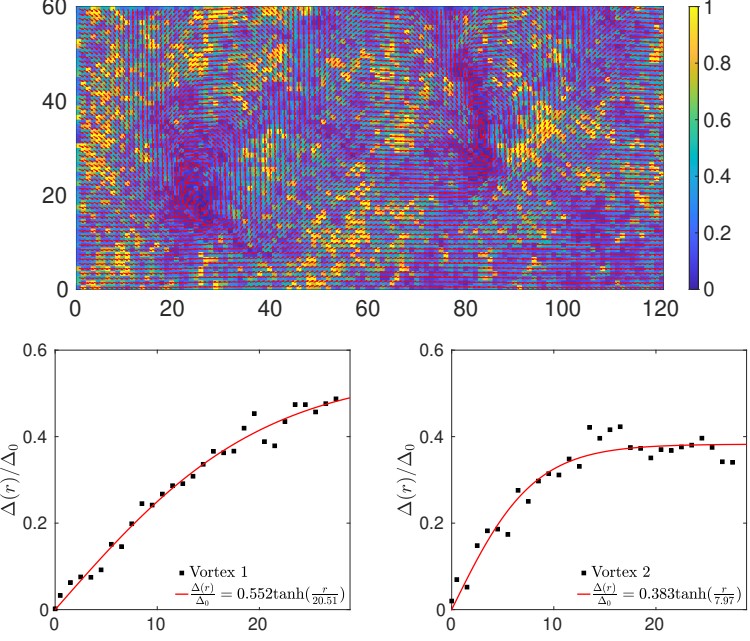

Figure 10: Upper: The spatial distribution of the order parameter and its phase in the intermediate disorder region $V = 1.5$. Lower: Vortex profile compared (red lines) with the Ginzburg-Landau prediction Eq. 4. Vortex 1 stands for the left vortex, and the right vortex is Vortex 2. The other parameters are $\langle n \rangle = 0.875$ and a lattice size $60 \times 120$. We employ a weaker coupling $|U| = 1.0$ so that the vortex is larger which facilitates the study of its core and highly deformed profile.

# 6 Characterization of the superconducting state in the presence of vortices and disorder

The results of previous sections suggest that while for weak disorder a vortex lattice is still formed, though with a different symmetry depending on the magnetic flux, the effect for stronger disorder is more drastic. No lattice structure or even short range position correlation can be discerned. It seems that vortices are located in regions with a very small value of the order parameter. Therefore, it is ultimately controlled by disorder, more specifically by the spatial inhomogeneities of the order parameter, and therefore not much influenced by the magnetic flux. In this section, we aim to understand in more detail to what extent disorder weakens the effects of the magnetic flux. We shall see that in certain cases it may even enhance superconductivity. We split the analysis of the interplay between disorder and magnetic flux in two parts. We first compute the covariance of the order parameter and the order parameter amplitude two-point correlation for a more quantitative assessment of the suppression of magnetic effects by disorder. In the second part, we show that in certain region of parameters, disorder increases experimental observables like the critical magnetic field and the spatial average of the order parameter.

## 6.1 Covariance and two-point correlation function of the order parameter amplitude

We compute the covariance of the order parameter with and without magnetic flux,

$$\text{cov}(\phi) = \langle (\Delta(0) - \bar{\Delta}(0))(\Delta(\phi) - \bar{\Delta}(\phi)) \rangle / \sigma(0)\sigma(\phi), \tag{5}$$

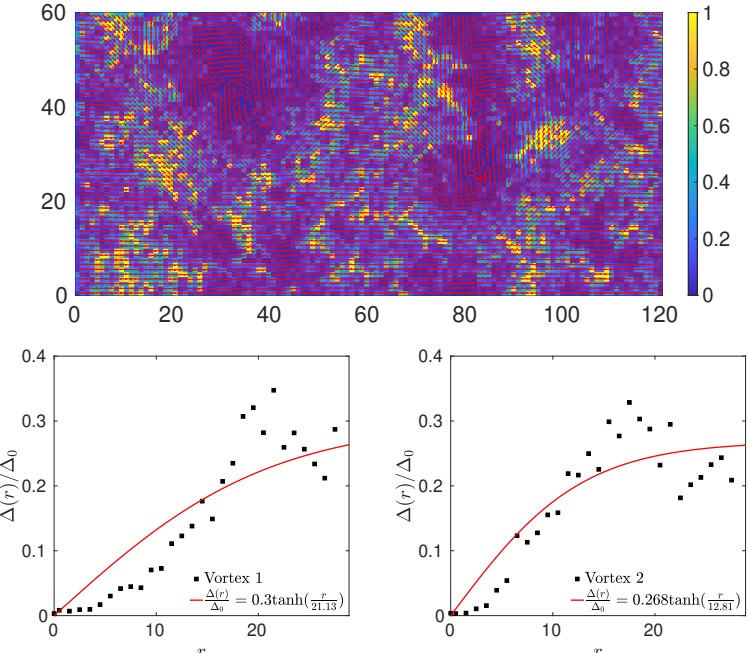

Figure 11: Upper: The spatial distribution of the order parameter and its phase in the stronger disorder region $V = 2.25$. Lower: spatial vortex profile compared with the Ginzburg-Landau prediction Eq. 4 in the clean case (red lines). Vortex 1 (2) stands for the left (right) vortex in the upper plot. The other parameters are $|U| = 1.0, \langle n \rangle = 0.875$ and a lattice size $60 \times 120$. Deviations from the theoretical prediction illustrate the important effect of disorder.

where $\sigma^2(0)$ is the variance of the order parameter without magnetic flux $\Delta(0)$, and $\sigma^2(\phi)$ is the variance of the order parameter in the presence of a magnetic flux $\phi$.

From previous results, we expect that for weak disorder cov$(\phi)$ is sensitive to $\phi$ because we observe different transitions in the vortex distribution. For strong disorder, the position of vortices does not change much with disorder, so we expect cov$(\phi)$ is only weakly dependent on $\phi$. Numerical results, depicted in Fig. 12, fully support these qualitative considerations. Even for $V = 0.5$, cov$(\phi)$ is relatively close to the one for weak fields $\phi/\phi_0 < 10$ because a vortex lattice is not yet formed so a flux has little effect. However, the formation of vortex lattice at $\phi/\phi_0 \sim 16$ reduces drastically the covariance. Disorder does not play an important role in this region. A further increase in the magnetic flux leads to changes in the vortex lattice structure and a further weakening of the correlations described by the covariance. The decrease rate of cov$(\phi)$ is reduced sharply around $\phi/\phi_0 \geq 24$ which is precisely the region where we stop observing any positional and orientational order in the vortex distribution. A magnetic flux above the critical one will eventually break down superconductivity. For stronger disorder, $V \geq 1.0$, the covariance decreases slowly with $\phi/\phi_0$ even for relatively large fields. Already for $V = 1.5$, which is still on the metallic side of the transition, the covariance is largely insensitive to $\phi/\phi_0$. This confirms that, in this region, vortex positions and the spatial distribution of the vortex core are closely related to the distribution of the superconducting order parameter and not to the strength of the magnetic flux, namely, the vortex is controlled by disorder and therefore the covariance does not change much.

We turn now to the two-point correlation function of the order parameter amplitude $\langle |\Delta(r)||\Delta(0)| \rangle$ which provides valuable information about the impact of a magnetic flux in a disordered superconductor. We note that we are not including phase fluctuations in our formalism so this observable provides only an upper bound for the loss of phase coherence. Since

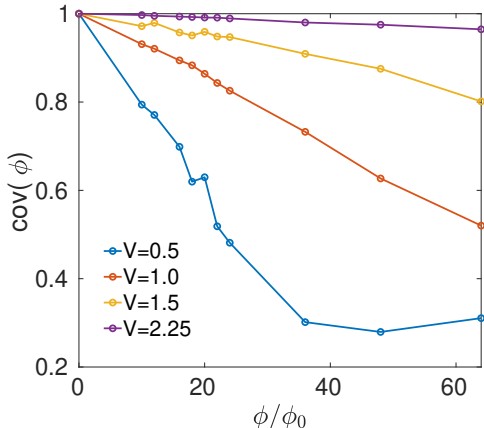

Figure 12: The dependence of the covariance cov($\phi$) Eq. (5) on the flux $\phi$ offers strong evidence that when disorder is weak $V = 0.5$, the impact of the magnetic flux is strong which is consistent with the observed different lattice distributions. However, for strong disorder, the effect of the flux is limited as vortices are located in regions where the order parameter is strongly suppressed by disorder effects. The flattening for large $\phi$ and weak disorder $V = 0.5$ signals the transition.

the disorder is not periodic in this study, we do not consider the periodicity when we calculate the correlation function, namely, we specifically consider only the sites that are at a specific distance, denoted as "$r$", from the chosen site "0", and then perform an average over all sites.

For weak disorder $V = 0.5$, see left plot of Fig. 13, we distinguish three different regions as $\phi/\phi_0$ increases: for $\phi/\phi_0 \leq 10$ the effect of the magnetic flux is small. We do not observe a decay of correlations. For $\phi/\phi_0 = 16, 20$, there exists a drop of correlations for long distances consistent with the formation of the vortex lattice. A further increase of the field results in a sharper drop of correlations consistent with the destruction of the vortex lattice. For stronger disorder $V \geq 1.5$, central and right plot of Fig. 13, the effect of the magnetic flux is relatively small which reinforces the idea that strong disorder suppresses the impact of the magnetic flux without necessarily breaking phase coherence.

## 6.2 Enhancement of the critical magnetic flux and the order parameter by disorder

An intriguing feature that we have observed is that the critical magnetic flux is enhanced by disorder. Results depicted in Appendix. C indicate that in the clean limit the maximum magnetic flux is $\phi/\phi_0 = 12$ for size $N = 60 \times 60$. However, see Fig. 4, even a weak disorder $V = 0.5$, enhances the critical maximum magnetic flux to $\phi/\phi_0 \sim 24$. In order to reach a more quantitative conclusion about whether the critical magnetic flux is enhanced by disorder, we determine this critical flux by both the study of the superfluid stiffness and a percolation analysis of the order parameter spatial distribution.

The superfluid stiffness $\frac{D_s}{\pi}$ presented in Fig. 14(a) is given by $\frac{D_s}{\pi} = \langle -k_x \rangle - \Lambda_{xx}(q, i\omega \to 0)$ [79], see Appendix. H for more details. In the weak disorder region, $D_s/\pi$ decreases sharply as the magnetic flux increases. In the intermediate disorder region, the superfluid stiffness decreases more slowly. $D_s/\pi$ is still finite even for $\phi/\phi_0 = 36$. This enhancement of the critical field is not monotonic. For a sufficiently strong disorder, Anderson localization effects trigger a transition even without a magnetic flux. This is illustrated for $V = 2.25$, at or very close to the transition, where the superfluid stiffness becomes compatible with zero for a much smaller field strength $\phi/\phi_0 = 16$. By compatible with zero we mean though the superfluid density

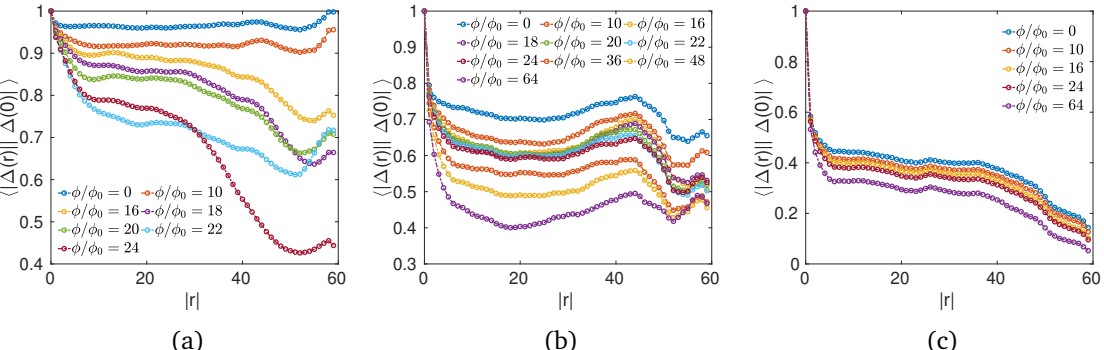

Figure 13: Two-point spatial correlation function of the order parameter. From left to right: $V = 0.5, 1.5$ and $2.25$. (a) The correlation is very sensitive to the magnetic flux strength. As a consequence, we observe substantial changes around the formation of the vortex lattice $\phi/\phi_0 = 16$ and its destruction $\phi/\phi_0 = 22$. (b) The breaking of positional order is signaled by the insensitivity of the correlation function to changes in the field around $\phi/\phi_0 \sim 20$. (c) Correlation function around the insulating transition. The dependence on the magnetic flux is rather weak in this region. Note the different range of fields in the left and right plots. The other parameters are $|U| = 1.25$, $\langle n \rangle = 0.875$ and the system size $N = 60 \times 60$.

is not zero its value is already very small so it is probably zero once quantum fluctuations, neglected in our mean field analysis, are considered.

We now proceed with another estimation of the critical field based on a percolation analysis of the order parameter spatial distribution. The percolation threshold for a 2D square lattice is $p_c = 0.59$ [80]. Results, depicted in Fig. 14(b), show that the critical flux for the breaking of superconductivity, within the metallic region, is enhanced by disorder. This is consistent with the previous superfluid stiffness analysis. Strictly speaking, the location of the transition depends on the cut-off value $\Delta_c$. Therefore, the percolation analysis gives only a rough estimation rather than a precise determination of the critical magnetic flux. However, in combination with the previous superfluid density results, it provides a consistent, albeit qualitative, picture of the role of disorder: up to intermediate strengths $V = 1.5$, disorder enhances the critical magnetic flux. A further increase of $V$, at or close to the insulating transition, leads to a suppression of the critical magnetic flux. The maximum enhancement occurs for intermediate values of the disorder strength $V \sim 1.5$.

We investigate now the effect of disorder on the spatial average of the order parameter and the spectral gap in the presence of a magnetic flux. For a fixed value of the disorder strength, see Fig. 15(b), the spatial average of the order parameter decreases as magnetic flux increases. However, the decrease is much slower as disorder is increased. Interestingly, we identify a region in the magnetic flux $\phi/\phi_0 \sim 20$ strength, close to the transition, where, for a fixed $\phi/\phi_0$, the spatial average of the order parameter is enhanced by disorder though it is still smaller than in the no disorder, no field limit. This is another example where disorder protects the superconducting state against magnetic effects that tend to weaken it. The average spectral gap, depicted in Fig. 15(a), shows qualitatively similar features though we could not clearly identify a region where disorder enhances it for a fixed magnetic flux.

In conclusion, even weak disorder debilitates magnetic effects in two dimensional superconductors. Ultimately, this is due to the fact that disorder makes the order parameter inhomogeneous in space. Quantum coherence effects, such as incipient localization or multifractality, tend to amplify this suppressing effect due to the enhancing of spatial inhomogeneities. We note that a microscopic model, as the one we employ, is necessary for a quantitative descriptions of this phenomenon.

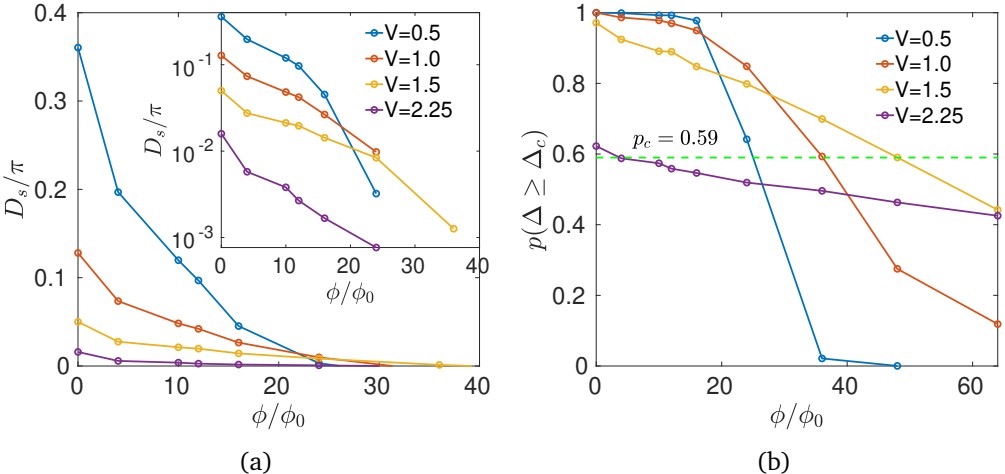

Figure 14: (a) The superfluid stiffness as a function of the magnetic flux. (b) The probability that $|\Delta(r)| \geq \Delta_c$, where $\Delta_c = 0.1\Delta_0$, and $\Delta_0 = 0.0897$. The crossing with $p_c = 0.59$ is the percolation prediction for the transition. The parameters are $N = 60 \times 60, |U| = 1.25$ and $\langle n \rangle = 0.875$.

# 7 Conclusion and Outlook

We have investigated the distribution of vortices in a two dimensional disordered superconductor by a completely microscopic approach based on the solution of BdG equations in the presence of a random potential and a magnetic flux introduced in the Peierls approximation. This is in contrast with most of previous calculations in the literature where the starting point is the semi-phenomenological Ginzburg Landau equations or by using the XY model and Monte-Carlo techniques. Until recently, our approach was not practically feasible because of limitations in the lattice size and therefore in the number of vortices that can be produced. Although limitations still exist, the rapid development of computational resources, and the use of state of the art numerical techniques, has made possible to obtain results for a range of parameters not far from the ones corresponding to weakly coupled metallic superconductors and to simulate a sufficient number of vortices to investigate different lattice configurations.

One of main results of this research is the observation, for a disorder strength not too strong, of different transitions in the vortex lattice as the flux strength increases. As was expected, for sufficiently weak disorder, a perturbed Abrikosov lattice is the configuration with lower energy. For a slightly stronger magnetic flux, the dominant configuration is instead a rectangular lattice. A further increase in the field strength leads to a phase characterized by short-range vortex repulsion but no clear evidence of Bragg's peaks which indicates loss of any discrete translation symmetry.

A further increase of disorder, or magnetic flux, still inside the superconducting side where global phase coherence holds, leads to the strong suppression of vortex repulsion. Indeed, the absence of vortex repulsion makes it at times difficult to distinguish individual vortices.

Another intriguing finding in this region is that the profile of single vortices is strongly deformed from the standard circular shape. Moreover, the vortex core becomes spatially inhomogeneous. It is plausible to expect that the vortex position and profile is mostly dictated by the spatial distribution of the order parameter rather than by the strength of the magnetic flux. As a result, the vortex distribution must be influenced by the multifractal-like properties of the spatial distribution of the order parameter [34,37,45,46]. However, larger sizes accommodating more vortices would be necessary to provide a more quantitative characterization.



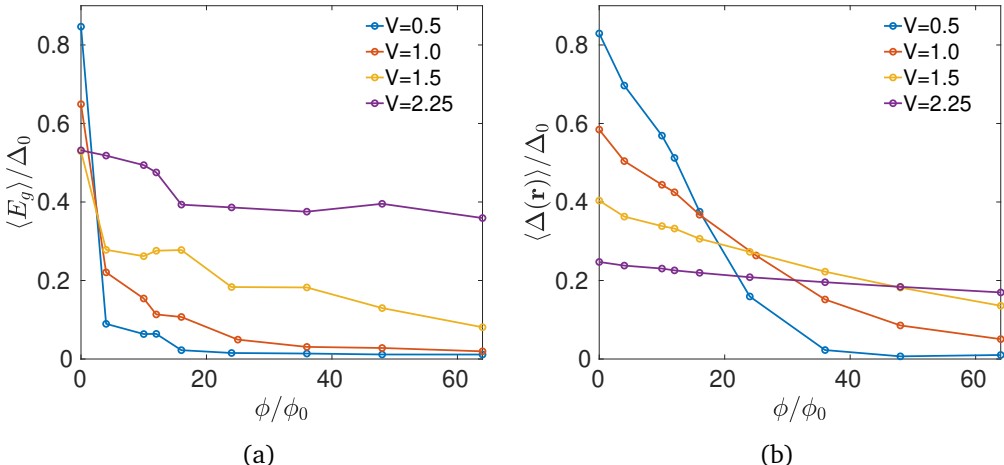

(a)                                    (b)

Figure 15: The spectral gap $\langle E_g \rangle$ (left) and the spatial average of the order parameter $\langle \Delta(r) \rangle$ (right) for $|U| = 1.25$ and $\langle n \rangle = 0.875$. While the spectral gap decreases monotonously with disorder and magnetic flux, we identify a region $\phi/\phi_0 \sim 20$ where $\langle \Delta(r) \rangle$ increases with disorder though its value is still smaller than $\Delta_0$, the order parameter in the absence of disorder and at zero field.

We also study the robustness of the superconducting state to the presence of vortices and disorder. A major result of this investigation is the observation of global phase coherence signaling a zero resistance state not only in the Abrikosov and rectangular lattice phase but also in the vortex repulsion phase provided that disorder or magnetic flux strength are not too strong. We have also identified a region of disorder close to the transition where the critical magnetic flux is substantially enhanced with respect to the clean limit. Likewise, for a fixed, and sufficiently strong field, $\phi/\phi_0 \sim 20$, we found that the spatial average of the order parameter is enhanced by disorder. However, it is still smaller than in the limit of no disorder and no magnetic flux.

Natural extensions of this work include the study of finite temperature effects and a more quantitative characterization of the vortex repulsion phase and the spatial deformation of the vortex core and profile, especially its relation to the multifractal-like spatial distribution of the order parameter. Another problem that deserves further attention is that of the interplay of magnetic effects in granular materials modeled by Josephson junctions nano-arrays where the superconducting state is also spatially inhomogeneous due to quantum coherence effects induced by variations in the grain size. It would also be worthwhile to investigate the vortex distribution in Hofstadter [69] superconductors and disordered multi-band topological superconductors where more stable vortex lattice configurations may exist.

# Acknowledgments

We thank illuminating conversations with Pedro Sacramento that helped improve the manuscript.

**Funding information** We acknowledge financial support from a Shanghai talent program and from the National Key R&D Program of China (Project ID: 2019YFA0308603). Bo Fan acknowledges support from the China Postdoctoral Science Foundation (Grants 2023M732256 and 2023T160409).

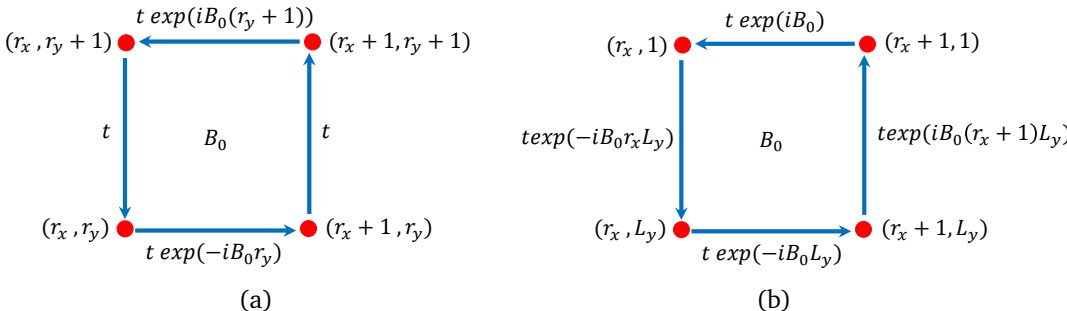

Figure 16: (a). The hopping term $t_{ij}$ at the lattice sites which are not the bottom and top boundaries. (b). The hopping between the bottom and top boundaries.

## A  The boundary conditions in the presence of the magnetic flux

In this appendix, we introduce in detail how the periodic boundary conditions are modified in the presence of magnetic flux, also see Ref [71, 72]. Although the periodic boundary condition which means that the boundaries are connected is still implemented to the lattice, it is no doubt that in the presence of vector potential $A = (-B_0 y, 0, 0)$, where $B_0 = \phi/(L_x \times L_y)$, the order parameter is no longer periodic. Here, we use the subscript $x$ and $y$ to specify the $x$ and $y$ direction for clarification.

As introduced in Section 2, the effect of a perpendicular magnetic field is introduced by Peierls substitution, which leads to $t_{r,r+\delta_y} = t$ and $t_{r,r+\delta_x} = t\exp(-ir_y B_0)$, where $\delta_x$ and $\delta_y$ are the nearest neighboring sites of $r$ along the $x$ and $y$ direction, which is illustrated clearly in Figure. 16(a). However, when the sites are in the bottom or top boundary, we need to introduce the extra phase along $y-$direction, see Figure. 16(b), to make sure that the sum of the phase in a minimum loop is still $B_0$. By considering all of this, the accumulated phase will be $L_x \times L_y \times B_0 = \pi\phi/\phi_0$, which means there will be $\phi/\phi_0$ vortices in the sample. Moreover, in order the wavefunctions are single valued, the accumulated phase must satisfy $\exp(i\pi\phi/\phi_0) = 1$, which implies that $\phi/\phi_0$ must be even. When this quasi-periodic boundary conditions are implemented properly, the magnetic translation symmetry is restored, which means in our system, the order parameter should follow the translation property:

$$\begin{cases} \Delta(r_x, r_y + L_y) = \Delta(r_x, r_y) \exp\left(i2\pi \frac{\phi}{\phi_0} \frac{r_x}{L_x}\right), \\ \Delta(r_x + L_x, r_y) = \Delta(r_x, r_y). \end{cases} \tag{A.1}$$

## B  Definition of the closed path and the position of vortices

After solving self-consistently the BdG equations (2) in the presence of magnetic flux, we obtain the on-site complex order parameter, which can be written as $\Delta(r) = |\Delta(r_i)|e^{i\theta_i}$. We can therefore separate the amplitude $|\Delta(r_i)|$ and phase $\theta_i$. We use the spatial distribution of the $|\Delta(r_i)|$ and $\theta_i$ to define the position of the vortices. It is known that around the vortex region, superconductivity is suppressed. The order parameter is almost zero at the vortex core. Therefore, it is better to first find those sites with order parameter smaller than a threshold value $0.2\Delta_0$. When the disorder is not that strong, those sites with heavily suppressed superconductivity possess a higher possibility to become a vortex. We define a closed path $\mathcal{L}$ around those positions. If the sum of the phase difference of two neighboring sites in this closed path $\mathcal{L}$ satisfies $\sum_{\mathcal{L}}(\theta_{i+\delta} - \theta_i) = \pm 2\pi$, this position corresponds to the center of a vortex core. In most cases, the numerical code finds easily the vortex by either considering the smallest loop, only four neighboring sites, around the point mentioned above, marked by the red arrow in



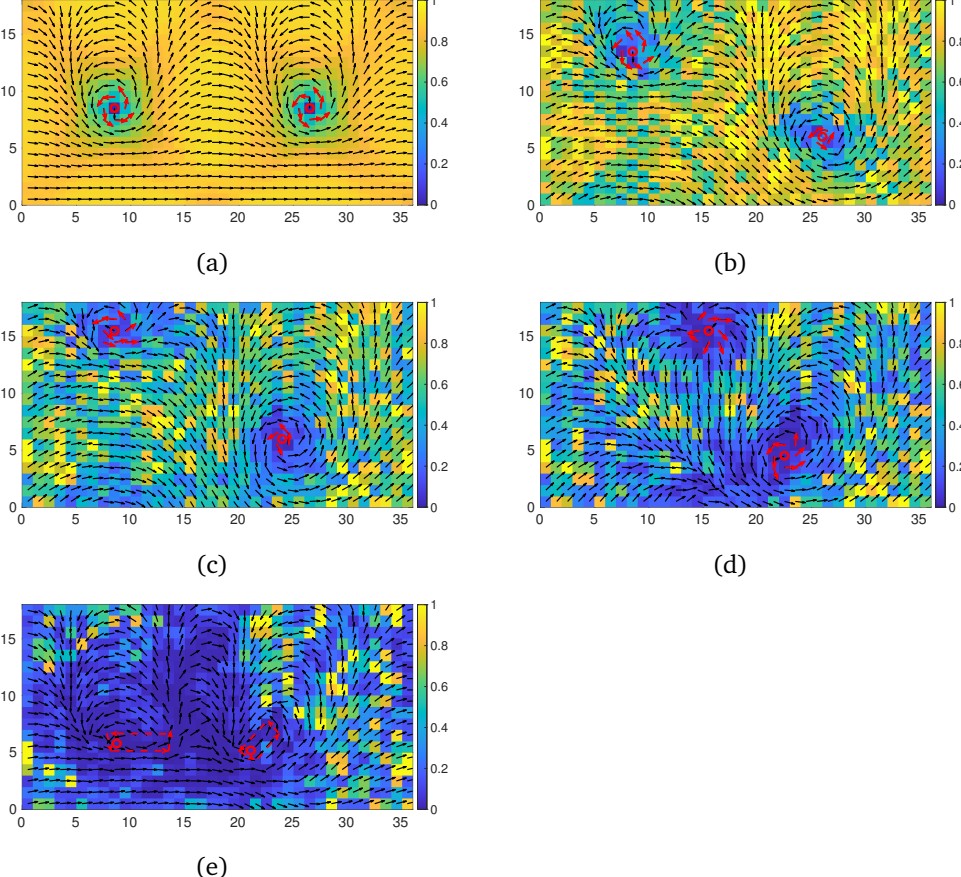

Figure 17: The spatial distribution of the amplitude of the order parameter $|\Delta(r_i)|$ (false color) normalized by its bulk value $\Delta_0 \sim 0.16$, and its phase $\theta_i$ (black arrows). The closed path is presented by the red arrows and the position of the vortex is marked by the red circle. The results are on a smaller system size $N = 18 \times 36$ in the presence of magnetic flux $\phi/\phi_0 = 2$, which means there have two vortices. The coupling constant is $U = -1.5$ and the average density is $\langle n \rangle = 0.875$. The disorder strength is $V = 0.0, 0.5, 1.0, 1.5$ and $2.25$ from (a) to (e).

Fig. 17(b) and 17(c) (the vortex on the right side), or the second smallest loop with eight sites, see Figs. 17(a) and 17(d). The vortex core is then located at the center of the corresponding closed path. However, some vortices cannot be identified by the code with the method mentioned above, especially in the stronger disordered region. Since we already know the number of vortices in the sample when we define the magnetic flux, that is $\phi/\phi_0$, we can find the rest of vortices by hand, as shown in Fig. 17(e). Although it is a much larger closed path, the sum of the phase difference along this path is still $2\pi$, if it contains a vortex.

As shown in the main text, at even stronger disorder or higher magnetic flux, it is difficult to identify the vortex by this simple way due to the strong spatial inhomogeneities of the order parameter. Although we couldn't identify vortices in some regions in the presence of high magnetic flux, we want to stress that there might contain one or more flux in these regions, which makes the ambiguous phase distribution, see Figure 18. More interestingly, in some regions, the phases at neighboring sites have opposite directions. Whether it is some kind of artifactual behavior, or novel physical mechanism still needs further studies. For that reason, we just simply plot the spatial distribution of the order parameter without presenting the position of the vortices to show the gradual suppression and eventual disappearance of superconductivity.

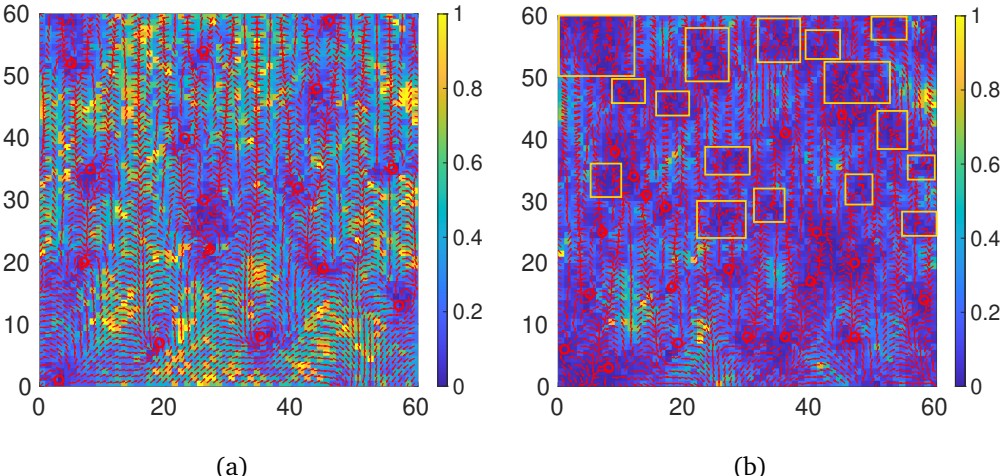

Figure 18: The spatial distribution of the order parameter amplitude $|\Delta(r_i)|$ (normalized by $\Delta_0 = 0.0894t$) and phase $\theta_i$ (red arrows) in the presence of magnetic flux $\phi/\phi_0 = 16$ (left) and $\phi/\phi_0 = 36$ (right). The red circles represent the position of the vortices core. Although the phases don't forms the standard vortex loop as we introduced earlier, we still expect the regions marked by yellow rectangles might contain one or more fluxes. The reason is that in these regions, the amplitude of the order parameter are highly suppressed by magnetic flux, and the phase distribution also show strange behaviors. The strength of random disorder is $V = 1.0$, and the other parameters are $|U| = 1.25, \langle n \rangle = 0.875$.

## C   The vortices distribution in the absence of disorder

In this Appendix, we present results of the vortices distribution in a finite size clean system in the presence of increasing magnetic fluxes, to show that how the vortices accommodate itself below the critical magnetic field. Since the size is finite and the system is symmetric, only configurations with a certain number of vortices respect the symmetry. Only when there are 12 vortices, a compressed Abrikosov lattice is reproduced. The order parameter also decreases fast with increasing magnetic field. Slightly increasing the magnetic flux further to $\phi/\phi_0 = 14$, the superconductivity breaks.

## D   Calculation of structure factor $S(q)$

The structure factor is a fundamental concept in the field of condensed matter physics and materials science, providing valuable insights into the arrangement and ordering of atoms within a lattice. It is also a type of Fourier analysis. In this particular study, we focus on investigating the vortex lattice and its properties. By analyzing the structure factor, we can gain insights into the distribution of vortices and the overall symmetry of the vortex lattice. Mathematically, the structure factor is defined as follow

$$S(q) = \frac{1}{\sum_{ij} f_v(r_i) f_v(r_j)} \sum_{ij} f_v(r_i) f_v(r_j) exp(iq(r_i - r_j)). \tag{D.1}$$

However, due to the spatial inhomogeneity of the superconducting order parameter, directly calculating the structure factor of the order parameter would introduce additional effects that obscure the information about the vortex lattice. To address this, we extract the vortex positions, denoting them as $f_v(r_i)$, and calculate the structure factor based on these positions.

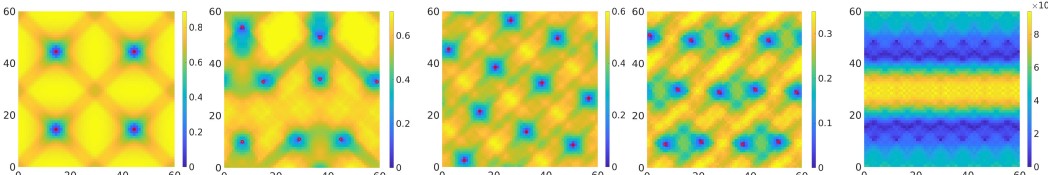

Figure 19: The spatial distribution of the order parameter amplitude $|\Delta(r_i)|$ (normalized by $\Delta_0 = 0.0894t$) in the presence of magnetic flux $\phi/\phi_0 = 4, 8, 10, 12$ and 14 from left to right in the clean limit. The red circles represent the position of the vortices core. The system size is $N = 60 \times 60$ and the other parameters are $|U| = 1.25, \langle n \rangle = 0.875$.

The essential steps are depicted in Figure 20. The pattern of structure factor is significantly improved, enabling a more accurate analysis of the vortex lattice.

## E   More results for the profile of vortices

Section 5 discusses the spatially inhomogeneous vortex core, and this appendix presents the additional results, see Fig. 21 for weak disorder $V = 0.5$ and Fig. 22 for stronger disorder $V = 1.5$. In the weak disorder $V = 0.5$, both the phase and the amplitude of the order parameter is closed to a circle, and its profile is well described by the GL theory. When disorder increases to $V = 1.5$, but still in the superconducting region, the shape of vortex core differ significantly from each other and it is never a circle. The vortex profile also deviate noticeably from the predictions of GL theory. However, by taking a sample average, as shown in Fig. 23, the rotational symmetry of the vortex core is restored, resulting in a standard circular vortex core that fits well with the GL theory. Moreover, the fitting parameter $r_0$, which characterizes the vortex size for the sample averaged vortex, decreases slightly with increasing disorder, indicating that the vortex core becomes smaller.

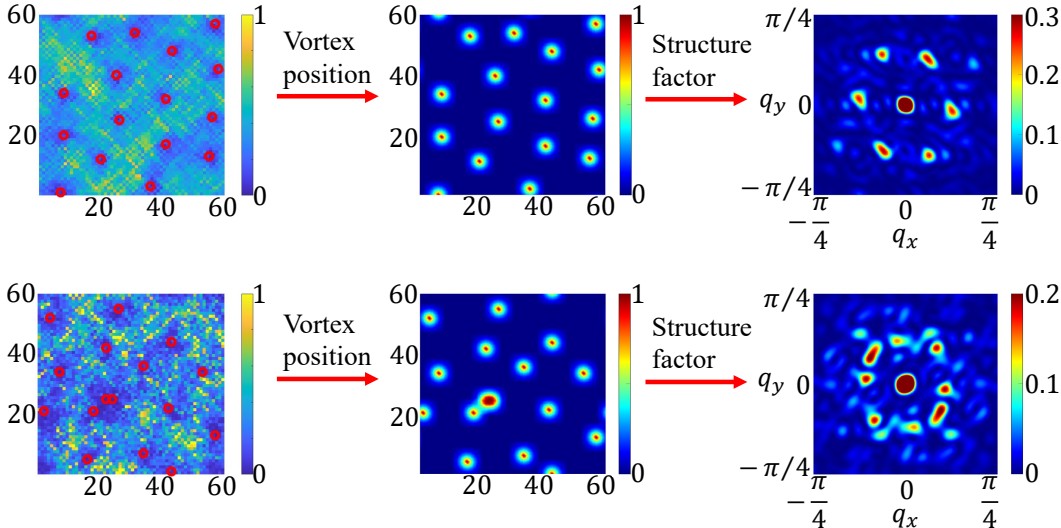

Figure 20: Process for obtaining the structure factor $S(q)$. The spatial distribution of the order parameter corresponds to the results in Figure 4. The upper panel demonstrates the sample with weak disorder $V = 0.5$ and the lower panel depicts intermediate disorder $V = 1.5$. The number of fluxes are fixed at $\phi/\phi_0 = 16$.

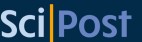

Figure 21: The spatial distribution of the order parameter and its phase in the weak disorder region $V = 0.5$, and the corresponding vortex profile with the GL fit Eq. 4. Vortex 1 means the left vortex, and the right vortex is Vortex 2. Five different disorder realizations are presented. The other parameters are $|U| = 1.0, \langle n \rangle = 0.875$.

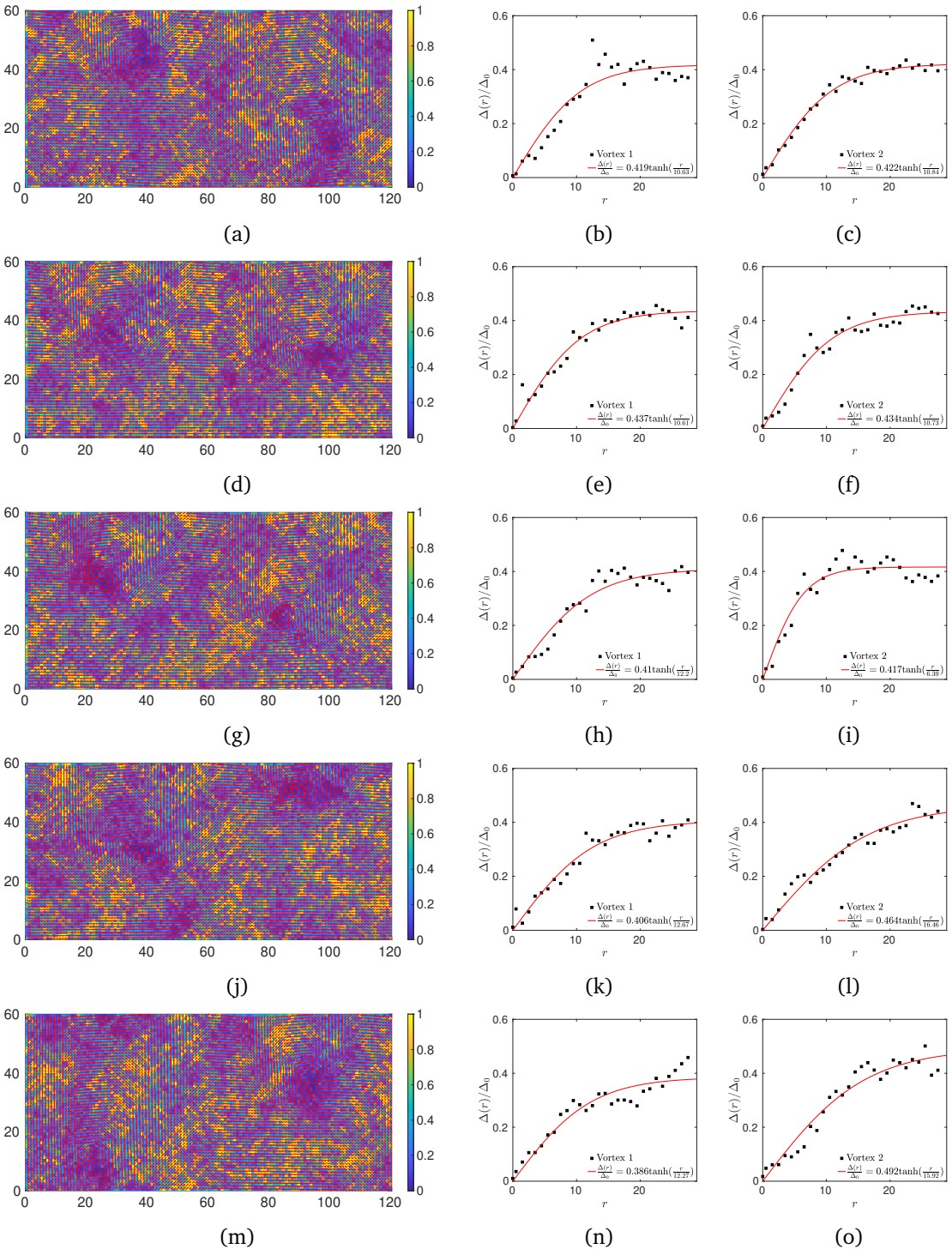

Figure 22: The spatial distribution of the order parameter and its phase in the intermediate disorder region $V = 1.5$, and the corresponding vortex profile with GL fit. Vortex 1 means the left vortex, and the right vortex is Vortex 2. Five different disorder configurations are presented. The other parameters are $|U| = 1.0$, $\langle n \rangle = 0.875$.

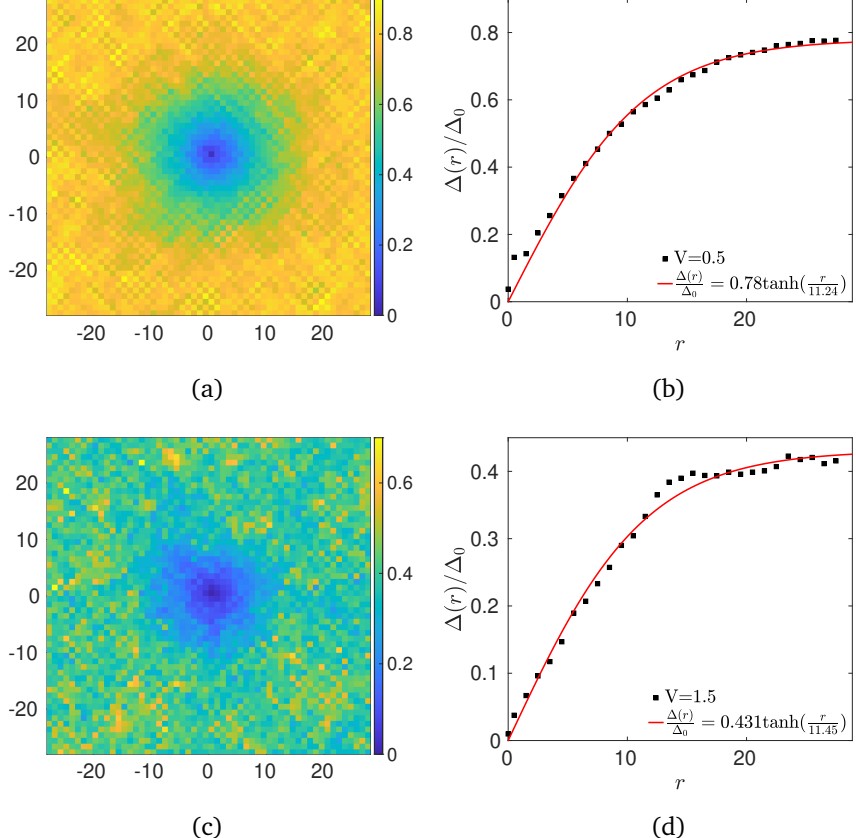

Figure 23: The sample average of the order parameter at the vortex region in the presence of weak disorder $V = 0.5$ (upper panel) and intermediate disorder $V = 1.5$ (lower panel), and the corresponding vortex profile with GL fit. For the weak disorder $V = 0.5$, we do sample average over 14 vortices. We calculate 24 vortices to do sample average for stronger disorder $V = 1.5$, so that we could remove the significant inhomogeneity in the order parameter amplitude. The other parameters are $|U| = 1.0, \langle n \rangle = 0.875$.

## F Confirmation of rectangular vortex lattice at a larger sample size

In this appendix, we provide further evidence of the existence of the rectangular vortex lattice by increasing the system size up to $L = 100$. Therefore, we will have more vortices in the sample which facilitates the analysis of its spatial distribution. It is important to stress that, in strictly two dimensions, due to localization for any disorder strength in the non-interacting limit, a larger system sizes effectively enhances the effect of disorder. For that reason, we will focus on this appendix on a weaker disorder strength $V = 0.25$ to be able to observe the triangular phase for small magnetic flux and the rectangular phase for stronger magnetic flux. We note that the rectangular phase is still observed for $V = 0.5$ which is the value chosen in the main text.

The vortex distribution in real and Fourier space for different values of the magnetic flux are depicted in Fig. 25 for $V = 0.25$ and in Fig. 26 for $V = 0.5$. For $V = 0.25$ and $\phi/\phi_0 = 20$, we observe a clear signal of a slightly deformed hexagonal lattice in Fourier space which corresponds with the Abrikosov triangular lattice in real space. However, for a larger magnetic flux $\phi/\phi_0 = 30$, the lattice distribution is fully consistent with a rectangular lattice. More

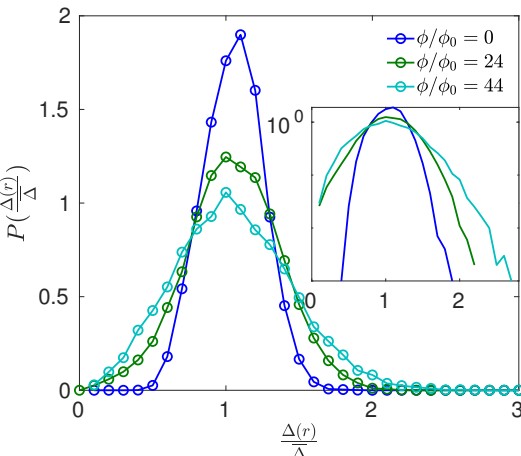

Figure 24: Distribution function of the spatial distribution of the order parameter for $V = 0.5$ and different values of the magnetic flux.

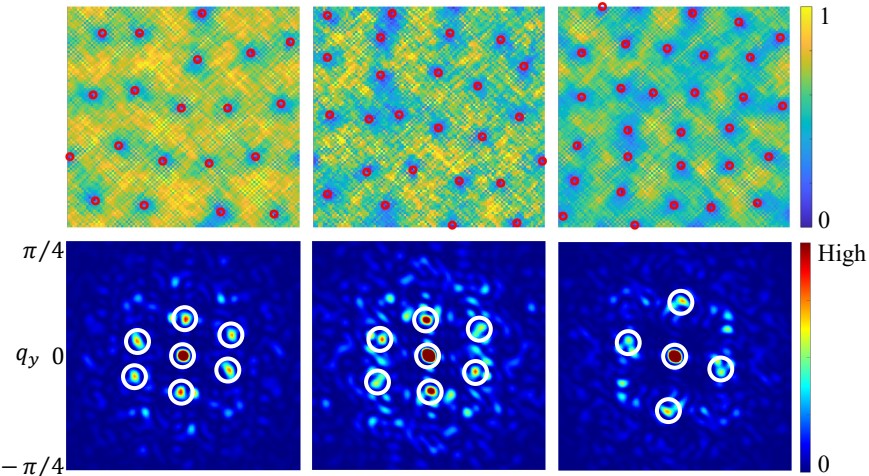

Figure 25: Top row: Spatial distribution of the vortex for $V = 0.25$ and, from left to right, a magnetic flux $\phi/\phi_0$ is 20, 24 and 30. Bottom row: Structure factor of the vortex distribution. The system size is $N = 100 \times 100$, and the other parameters are those of the main text, $U = -1.25$ and $\langle n \rangle = 0.875$.

specifically, the distribution seems to be sensitive to the microscopic details of the disordered potential. Depending on the disorder realization, vortices in some parts of the sample seems to start forming a triangular lattice while in other parts no such pattern is observed. Since Anderson localization in two dimension occurs for any disorder strength and the sample size is larger now $L = 100$, we expect stronger inhomogeneities for the same disorder strength. Stronger spatial inhomogeneities will eventually prevent the observation of the triangular phase that requires no or very weak disorder. It would be necessary larger sample sizes to clarify whether the transition between triangular and rectangular is sharp or it is just a crossover as a function of the field strength. However, we rule out any important role of multifractality or other direct precursor of localization. The distribution of the order parameter, see Fig. 24, in this range of parameters is Gaussian and not log-normal and level statistics, see appendix G follows closely the prediction of random matrix theory which is a clear signature that the system is deep in the disordered metallic phase.

For $V = 0.5$, we do not observe the standard triangular lattice because disorder is already too strong given the larger size. However, the rectangular lattice phase is clearly observed for a wide range of parameters between $30 \leq \phi/\phi_0 \leq 44$. In conclusion, the results for a larger size, $L = 100$ confirm the existence of the rectangular Bragg vortex lattice for weakly disordered two-dimensional superconductors in the presence of a perpendicular magnetic flux.

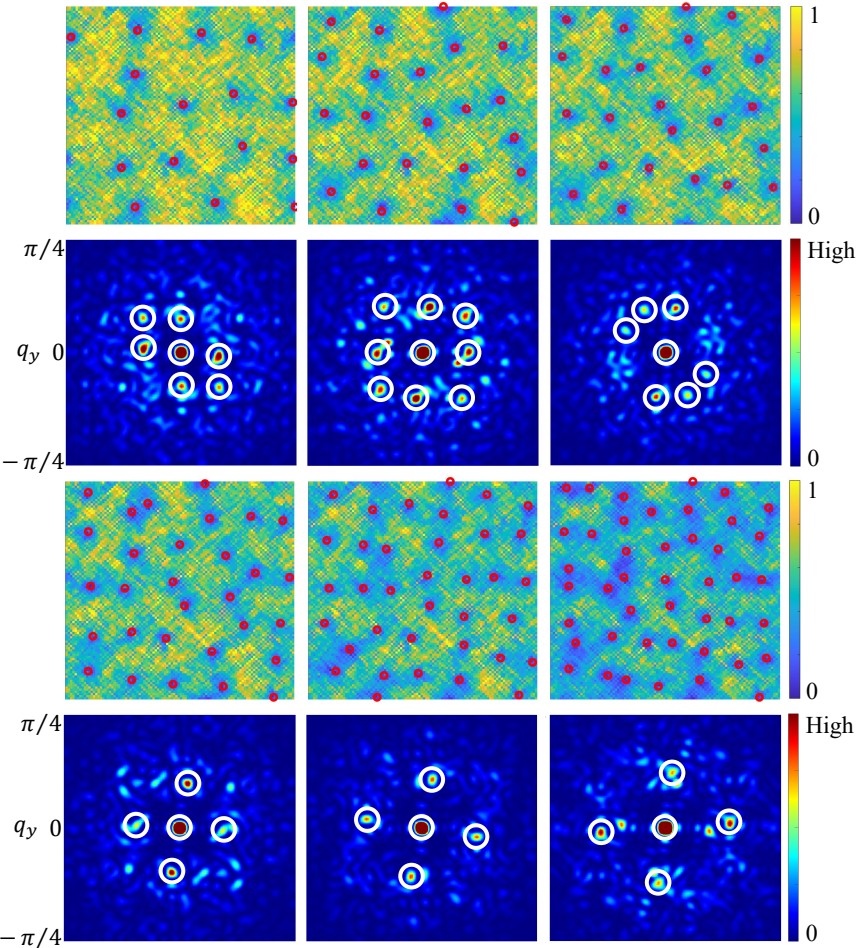

Figure 26: The spatial distribution of the vortex and its structure factor for $V = 0.5$ and a magnetic flux $\phi/\phi_0 = 16, 22, 24, 30, 36, 44$. The other parameters are the same as those of Fig. 25.

## G  Spectral analysis at weak disorder in the presence of magnetic flux

In this appendix, we study the level statistics of the eigenenergies of the BdG equations with the aim to clarify whether $V = 0.5$ is still in the weak disorder region where multifractal effects are expected to be negligible. For that purpose, we compare the level spacing distribution $P(s)$, the probability of having two eigenvalues at a distance $s$ in units of the local mean level spacing, with the Wigner-Dyson surmise which is a very good approximation of the random matrix prediction.

We note that without a magnetic flux $\phi/\phi_0 = 0$, the Hamiltonian is time reversal invariant and rotational symmetric. Therefore, we should compare our results with that of the Gaussian

orthogonal ensemble (GOE),

$$P_{GOE}(s) = \frac{\pi}{2}s \exp\left(-\frac{\pi}{4}s^2\right). \tag{G.1}$$

However, a magnetic field breaks time reversal symmetry so in that case the comparison should be with the Gaussian unitary ensemble (GUE),

$$P_{GUE}(s) = \frac{32}{\pi^2}s^2 \exp\left(-\frac{4}{\pi}s^2\right). \tag{G.2}$$

Results depicted in Fig. 27 confirm a good agreement with the random matrix prediction. This confirms both that localization effects are not important and that the magnetic flux breaks time reversal symmetry.

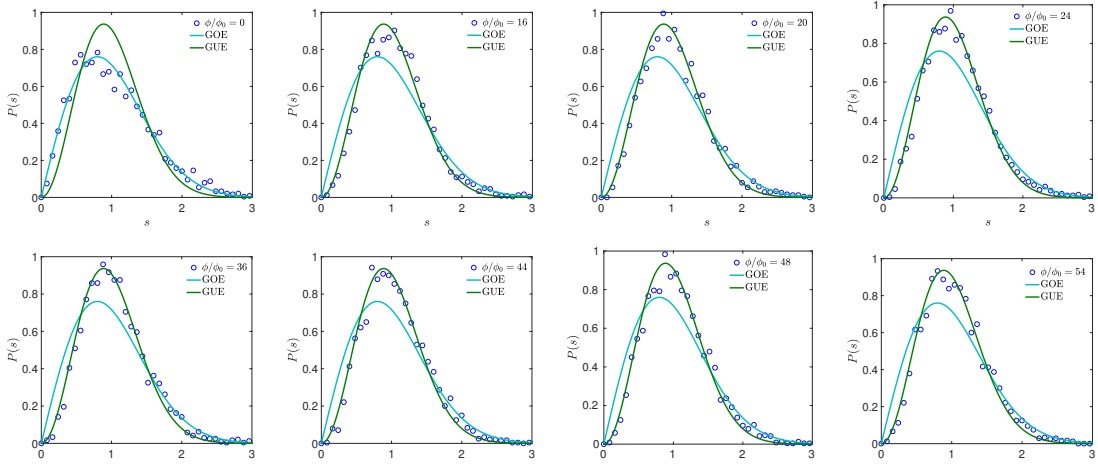

Figure 27: The level spacing distribution $P(s)$ for $V = 0.5$ and different external magnetic flux is $0, 16, 20, 24, 36, 44, 48$ and $54$. The GOE is described by Eq (G.1), and GUE is Eq (G.2). The system size is $N = 100 \times 100$, $U = -1.25$ and $\langle n \rangle = 0.875$. For the statistical analysis, we only take 3000, 30% of the total, eigenenergies around the band center $E = 0$. The observed good agreement with the random matrix prediction precludes any important effect of multifractality in this weakly disorder region.

# H Calculation of the superfluid stiffness $D_s/\pi$

In this Appendix, we present the detailed formulas to calculate the superfluid stiffness. Solving the self-consistent BdG equations, we will get the eigenvalues $\{E_n\}$ and the corresponding eigenvectors $\{u_n(i), v_n(i)\}$. With those outputs, we can calculate the superfluid stiffness $D_s/\pi = \langle -k_x \rangle - \Lambda_{xx}(q = 0, i\omega \to 0)$, where $\langle -k_x \rangle = \frac{2t}{N}\sum_i \sum_n [v_n(i)v_n^*(i+x) + v_n^*(i)v_n(i+x)]$ is the kinetic energy along the $x$ direction. The second term $\Lambda_{xx}(q = 0, i\omega \to 0)$ can be obtained from the bare current-current correlation function [48, 81, 82], which is given by

$$\begin{aligned}
\chi_{ij}(j^x, j^x, i\omega) = 2t^2 \sum_{nm} & \frac{u_n^*(i+\hat{x})v_m^*(i)(v_m(j+\hat{x})u_n(j) + v_n(j+\hat{x})u_m(j))}{\omega + i\eta + E_n + E_m} \\
& - \frac{v_n(i+\hat{x})u_m(i)(u_m^*(j+\hat{x})v_n^*(j) + u_n^*(j+\hat{x})v_m^*(j))}{\omega + i\eta - E_n - E_m} \\
& - (j+\hat{x} \leftrightarrow j) - (i+\hat{x} \leftrightarrow i) + (i+\hat{x} \leftrightarrow i, j+\hat{x} \leftrightarrow j),
\end{aligned} \tag{H.1}$$

where $(j+\hat{x} \leftrightarrow j)$ means swapping the site index $j+\hat{x}$ and $j$ in the presented expression. We then obtain $\Lambda_{xx}(q = 0, i\omega \to 0) = \frac{1}{N}\sum_{ij} \chi_{ij}(j^x, j^x, i\omega \to 0)$ by summing over all sites $i$ and $j$.

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
