# Peer review of "Exploring the vortex phase diagram of Bogoliubov-de Gennes disordered superconductors"

_SciPost Physics, doi:SciPost Phys. 15, 196 (2023)_

## Round 1 · Referee Report · Anonymous (Referee 1) · 2023-7-17

Report
This work presents a detailed study of the vortex phase diagram of a conventional type-II superconductor. Due to the sizes considered, the self-consistent nature of the approach and physical quantities studied, it provides a significant improvement on previous results and identifies and characterizes different phases. Particular attention is paid on the effects of magnetic flux and disorder on the changes in the vortex lattice structure and the distortions on the vortices. The work is an interesting addition to the understanding of a disordered superconductor in a magnetic field and should be published.
Questions/remarks:
1- Can the authors estimate for which size do the results improve and differ significantly from previous results in the literature? What is more relevant to the improvement of this study: larger systems or the self-consistency+ study of lattice deformation, etc?
2- The authors only consider the effect of the vector potential and neglect (as is standard practice) the effect of Zeeman term (coupling of the spins to the external magnetic field). If the magnetic fields are large, the spin coupling may have a significant effect, unless, for instance, the g-factor is small. It would be worthwhile to comment on the approximation of ignoring the Zeeman coupling.
3- How is the stiffness calculated explicitly in magnetic field? It would be useful if some detail of the calculation is presented.
4- Since the authors are able to work with larger systems, it would be interesting, in some future work, to consider gapless systems, such as d-wave superconductors.
Author: Antonio Miguel García García on 2023-09-10 [id 3969]
(in reply to Report 1 on 2023-07-17)See attached file for a detailed response to the referee report
Author: Antonio Miguel García García on 2023-09-10 [id 3971]
(in reply to Report 3 on 2023-07-27)See attached file
Attachment:
replyreferee3.pdf

---

## Round 1 · Referee Report · Anonymous (Referee 2) · 2023-7-19

Strengths
1) Methodology: Authors use a microscopic model to investigate the interplay between disorder and vortex formation. For strong disorder this is superior to 'conventional' phenomenological approaches.
Considered system sizes are much larger than in previous investigations.
2) Careful and physically sound discussion on the various aspects of their findings.
3) Good introduction into the subject.
Weaknesses
1) At few places it is hard to correlate the discussion with the results shown in the figures.
2) Figure labeling
Report
In this paper authors investigate the interplay of
disorder and vortex formation on the basis of an
attractive Hubbard model with on-site disorder which is coupled to a magnetic field and solved within a Bogoliubov-de Gennes approach.
Different regimes in the field-disorder phase space are identified. These comprise the conventional Abrikosov lattice in the small disorder regime, the transition toward a rectangular lattice at 'intermediate' fields, and the loss of translational invariance at even higher fields. Also the superconducting properties as a function
of the field are studied where it is found that up to intermediate disorder strengths the critical magnetic flux is enhanced. Moreover, for large magnetic fluxes disorder can even enhance the average superconducting order parameter.
This is an interesting paper which provides new insight into the actual and complex problem which makes a step forward to understand the influence of disorder on the vortex formation in superconductors. The paper is well written and meets the criteria for publication in SciPost.
I therefore recommend publication of the manuscript in SciPost after the points in "Requested changes" have been considered.
Requested changes
1.) According to Abrikosov theory the 'size' of the vortex core is determined by the coherence length. Despite that it is a central quantity in vortex
physics the term 'coherence length' appears only once in the caption to Fig. 1. In my opinion it should be straightforward to evaluate the coherence length as a function of disorder (e.g. from the current-current correlations) and then compare with the vortex profile shown in Figs. 7-10.
2.) For the clean system the vortex lattice is only shown for values of the flux up to \phi/\phi_0=18. It would strengthen the discussion when authors would add to Fig. 3 a row with V=0. In fact, Fig. 1 seems to
indicate that there is also a transition to a rectangular structure for V=0 whereas on page 14 (2nd row) it is claimed that this structure results
from a compromise between disorder and magnetic flux. The question is therefore, whether for the clean case the lattice stays triangular up to high fields.
3.) Page 12, last paragraph: The quantity \xi_0 is introduced as the vortex separation in the clean limit. I don't understand this definition because the vortex separation should depend on the flux. Does \xi_0=12 refer to the same flux where the rectangular lattice is
observed? Please clarify!
4.) page 17, 2nd paragraph: "It is expected that the profile of the order parameter should match with the magnetic field inside the vortex.....". This statement and the following is misleading. The profile of the order parameter is determined by the coherence length whereas the decay of the magnetic field is ruled by the penetration depth and the functional forms of both quantities do not necessarily coincide. Eq. 4 is rather an Ansatz which allows to fit the order parameter profile but I would not relate this to a functional form for the magnetic field.
5.) page 16: The correct limits for the definition of the superfluid stiffness are \omega=0 and the transverse momentum q->0. At the end of the same paragraph it is stated that (for V=2.25) "the superfluid stiffness
becomes zero for a much smaller field strength \phi/\phi_0=16." However, Fig. 13a still reveals a finite stiffness of D_s=10^-3 - 10^-2 for this value of the flux.
6.) Fig. 12: Why the correlation function is not periodic? Does the plotted r-range correspond to half of the lattice size?
Minor issues:
a) Eq. 1: Either the hamiltonian is defined for arbitrary hopping parameters, then one should replace -t-> t_{ij}. Or one introduces nearest-neighbor hopping from the beginning. Then this should be indicated in the sum over "i" and "j".
b) Eq. 3: Replace t_{ij} -> t_{i,i+\delta} and put it under the sum.
c) The results in Sec. IV are for 60x60 lattices. This is only specified at the end of Sec. IV but should be already defined at the beginning.
d) In all figures which report the Fourier transform the range of momenta should be indicated.
Author: Antonio Miguel García García on 2023-09-10 [id 3970]
(in reply to Report 2 on 2023-07-19)See attached file
Attachment:

---

## Round 1 · Referee Report · Anonymous (Referee 3) · 2023-7-27

Strengths
1. Results are sound and interesting, and may open a new pathways for future research
2. Results were obtained in a novel fashion
3. Sufficient details are presented
4. Mostly clear presentation
Weaknesses
1. Some assumptions and parameter values are not stated clearly and should be justified more, especially the uniform filed approximation.
2. One of these assumptions is the form of the order parameter around the vortex under Eq. (4), which is at odds with the well-known Abrikosov vortex solution in the Ginzburg-Landau theory, Δ(r)~tanh(r).
3. Organizational issues: paper somewhat lacks focus. For example, introduction refers to Appendix figures multiple times.
4. Least important, but grammar and sentence structure could generally be improved throughout the text.
Report
With some necessary changes I believe this manuscript may in principle be acceptable for publication in SciPost. Specifically I believe this research can open a new pathway for further follow-up work. The authors have carried out a numerical study of vortex lattices in a square lattice superconductor in the presence of disorder by solving real-space BdG equations for a previously inaccessible lattice size. One of the main results is an observation of a phase transition from the usual triangular vortex lattice to a square lattice as the magnetic field increases.
There are several issues the authors should address, which I think can significantly improve the manuscript. First, the Hamiltonian studied in this paper, Eq. (1), is the Harper-Hofstadter model with addition of disorder and interactions. Superconductivity has been studied in this model in the absence of disorder in some earlier studies that the authors should cite. More importantly, the model assumes that the uniform magnetic field fully penetrated the superconductor, which is a valid approximation only close to a second order phase transition. This assumption should be made more explicit. Related to this, there is band reconstruction in the Hofstadter model and the electrons form Hofstadter bands separated by gaps. In the regime considered by the authors these are essentially Landau levels. Another important assumption the authors appear to make is that the superconducting gap Δ is much larger than the separation between the Hofstadter bands/Landau levels, since U is taken to be on the order of the full bandwidth at zero field. Is this correct? If so, this should be stated. Reentrant superconductivity has been predicted when this assumption no longer holds (see Rev. Mod. Phys. 64, 709 and references therein), a potential direction for future research that may be relevant to mention.
It may also be worthwhile to note that in the Hofstadter model in the Landau gauge the unit cell is extended into the magnetic unit cell, which for the parameter values used would be of length at least 625 if I am not mistaken. This is much larger than the lattice size used in the numerical calculations, which means that the true Hofstadter regime is likely not being accessed.
Second, and related to this, in the introduction the authors state that they studied 100x100 lattices, but later a 60x60 lattice is used for some parameter values. This should be mentioned earlier in the paper. Similarly, ranges of other parameter values like U and V should be mentioned earlier, and when different values are used some explanation should be given as to why the particular values were chosen.
Third, the most interesting finding in the paper is probably the phase transition between the triangular and square lattices. It is stated that no transition occurs in the absence of disorder, but numerical data is not presented to support this claim. Could the authors provide this data for completeness?
Fourth, concerning the vortex profile study, I am not sure about the validity of the form of Δ(r) used by the authors as stated below Eq. (4), at least for weak disorder. It is well-known that in the absence of disorder the profile is Δ(r)~ tanh(r) in the Ginzburg-Landau theory, and this seems to fit the numerical data better than some of the fits in Figures 7-9. Also, the parameter Δ_0 is not defined, and I would suggest changing A and B parameters to something like a and b, to avoid confusion with the vector potential and magnetic field. In general, there is an apparent contradiction in introducing Eq. (4) as the magnetic field is assumed to be uniform in the model studied by the authors; this should be explained more clearly. In this section |U|=1 is used instead of 1.25 used in the previous section, is there any particular reason for this? Finally, in Fig. 10 some sharp peaks in Δ are seen around r=15 and -20, do the authors have any explanation for their origin?
Fifth, concerning the study of correlations in Sec. VI, one question that does not appear to be addressed in the correlation between the disorder potential V_i and the gap function distribution. I think the authors should present some data to address this issue. Also, concerning Fig. 13a, it is stated that D_s becomes zero for V=2.25, but this appears at odds with the insert in the figure.
Sixth, in Fig. 18 and 19 in Appendix C, are (a), (d), (g), (j) and (m) different disorder realizations?
Seventh, in terms of organization I wonder if section III should be simply incorporated into section IV. Since figures from appendices are referred to extensively in the introduction, it also seems appropriate to move them into the main text, along with any relevant explanations. Alternatively, they should not be referred to in the introduction.
Finally, though this is not a huge issue I think the grammar can be improved significantly in the text. For example, on page 5, paragraph 1: “it has also been identified a range of parameters where...” is incorrect grammar. It should read instead "A range of parameters has been identified where..." I saw multiple errors of this form throughout the text, in addition to other typos. The authors also refer to 'vortices overlap' several times, which should instead read 'vortex overlap.' These should be easy issues to address.
Overall, I think the manuscript is interesting and will meet the criteria for publication once these issues are addressed.
Requested changes
1. Identify Eq. (1) as the Harper-Hofstadter model and state the assumptions behind it more explicitly, as well as clarify the parameter regime being studied.
2. In general, state which ranges of parameters were used and why earlier in the text.
3. Present data showing no phase transition between triangular and square lattices occurs in the absence of disorder.
4. Justify the use of Eq. (4) better, and why it is assumed that Δ(r) is not linear in r for small r at weak disorder.
5. Study the correlation between the disorder potential V_i and the gap function distribution, as several claims appear to be made about it without supporting evidence.
6. Clarify captions in Fig. 18 and 19 in appendix C.
7. Move figures from appendix to main text, or change discussion in the introduction. Possibly move section III into section IV as it is too short.
8. Proofread the manuscript carefully and fix typos and grammar issues.

---

## Round 2 · Referee Report · Anonymous (Referee 1) · 2023-9-11

Report

In my view the authors have answered the comments and suggestions and the new version is ready for publication.
  • validity: -
  • significance: -
  • originality: -
  • clarity: -
  • formatting: -
  • grammar: -

Author:  Antonio Miguel García García  on 2023-09-27  [id 4016]

(in reply to Report 1 on 2023-09-11)

We thank the referee for accepting the paper for publication and the previous comments and suggestions that helped make the paper better.

---

## Round 2 · Referee Report · Anonymous (Referee 2) · 2023-9-12

Strengths

See previous report

Weaknesses

due to revisions previous weaknesses have been mostly removed but one, see report

Report

Authors have convincingly replied to the issues I have raised in my previous report and have revised the manuscript accordingly. There is, however, still one minor point related to the correlation function in Fig. 13. While authors have provided a better definition of the correlation function they should specify the system size in the caption to Fig. 13. For a NxN lattice the maximum distance is N/sqrt(2) along the diagonal. For a 60x60 lattice this would correspond to a maximum distance of ~42 and the correlation function for larger distances, i.e. 42+d, should be the same than for 42-d. This is what I meant with 'periodicity' in my previous report. So I guess that authors have used probably 100x100 for Fig. 13 but this should be specified.

Requested changes

  • Provide the system size in the caption to Fig. 13

  • validity: high
  • significance: high
  • originality: high
  • clarity: high
  • formatting: good
  • grammar: good

Author:  Antonio Miguel García García  on 2023-09-27  [id 4017]

(in reply to Report 2 on 2023-09-12)

See attached file with a detailed response to the referee comments and questions.

Attachment:

response_report2.pdf

---

## Round 2 · Referee Report · Anonymous (Referee 3) · 2023-9-26

Report

The authors have addressed of all my main comments. Concerning point 5, to clarify, what I meant is the correlation between the spatial distribution of the SC gap Δ and the disorder potential, for a fixed realization of the disorder. One expects that, as the authors claim, that the vortex cores are pinned to areas with strongest disorder. Figure R4 seems to suggest that this is indeed the case, at least for stronger disorder, but it is not clear from data presented in the manuscript. A plot of V(r) alongside the plots of Δ(r) already in the text would address my question. This seems especially important given the discussion by the authors of the need for a self-consistent BdG calculation, since they state that the disorder in Δ(r) does not follow the same distribution as the disorder in V(r).

Additionally, I would suggest another proof-reading of the text as many grammatical errors remain. For example, on page 15: "where it is observed a clear deformation" should instead read "where a clear deformation is observed." Another common mistake is the use of "the vortices position," which should either read "the vortex position" or "the position of vortices." Once these are fixed, I believe the manuscript will be acceptable for publication.
  • validity: -
  • significance: -
  • originality: -
  • clarity: -
  • formatting: -
  • grammar: -

Author:  Antonio Miguel García García  on 2023-09-27  [id 4018]

(in reply to Report 3 on 2023-09-26)
Category:
answer to question

See attached file with a detailed response to the referee comments and questions.

Attachment:

response_report3.pdf

---

## Round 2 · Author Response

Dear Editor,
Thanks for forwarding the three referee reports. A detailed response has been submitted separately. We think we have addressed all referee comments and suggestions satisfactorily. Indeed, the three reports were constructive and have helped make the paper better.
We are enclosing an updated manuscript.
We hope that the paper is now suitable for SciPost.
Your Sincerely,
The authors.

---

## Round 2 · List of Changes

Following the referees comments and suggestions, we have made the following changes in no particular order:
1. We have carried out a careful profreading of the full manuscript.
2. The plots providence direct evidence of the existence of triangular vortex lattice in the clean limit have been moved from the appendix to section III of the main text.
3. In section V, we switched to the Ginzburg-Landau theory prediction of the profile of the order parameter inside the vortex to compare with the numerical result from the solutiosn of the BdG equations.
4. In section II, after the introduction of the model, we have clarified why the range of parameters we are interested in are quite different to those employed to describe the physics of Hofstedter superconductors.
5. We have added structure factor plots (Fig. 2, 3, 5, 7) in order to provide sharop evidence of the vortex latice structure. Details of the calculation are in appendix D.

---

## Round 3 · Author Response

Dear Editor,
we have addressed all referee comments and have updated the manuscript accordingly.
We hope you find the paper suitable for publication.
Yours Sincerely,
Bo Fan and Antonio Garcia

---

## Round 3 · List of Changes

1. We have carried out a careful additional proofreading of the paper.
  2. In page 9, when we comment on Fig. 4, we have added details about the relation between the spatial distribution of the disordered potential and that of the order parameter.
  3. We have added the lattice size in Fig. 13

---

## Editorial Decision

published